

# Benchmarking Photolysis Rates: Species for Earth and Exoplanets

Sophia M. Adams[1], James Manners[2], Nathan Mayne[1], Mei Ting Mak (麥鎂婷)[1], and Éric Hébrard[1]

[1]Department of Physics and Astronomy, Faculty of Environment, Science and Economy, University of Exeter, Exeter EX4 4QL, UK
[2]Met Office, Fitzroy Road, Exeter EX1 3PB, UK

**Correspondence:** Sophia Adams (sa1076@exeter.ac.uk)

**Abstract.** Using the Socrates photolysis scheme, we present newly calculated photolysis rates under modern Earth atmospheric conditions for species directly relevant to Earth and species relevant to different atmospheric compositions. We compare to a previous photolysis comparison exercise, namely PhotoComp 2011. Overall, we find good agreement between our results and previous work, with discrepancies usually caused by the implementation of temperature dependent cross-sections or quantum

yields and updated or higher resolution input data. We provide a new set of benchmark photolysis rates for additional species both for Solar irradiance and when irradiated by an M dwarf host star. In general, the higher actinic flux at far-UV and shorter wavelengths of the M dwarf compared to the Sun drives increased photolysis rates for reactions with high threshold energies. This work provides an updated set of benchmark results for further studies of photolysis in the Earth's atmosphere and that of other planets.

## 1   Introduction

Photochemistry is chemistry driven directly by light, and, in the context of studying planetary atmospheres, by the stellar irradiation. High energy photons, typically within the UV wavelength range, can break down molecules in the upper atmosphere and initiate various chemical reactions and pathways. This process, photolysis, is the degradation of a reactant molecule into constituent product atoms or molecules initiated by the absorption of a photon. Photochemistry plays an important role in the atmospheres of Earth, both modern and early, and potentially for exoplanets (planets orbiting stars other than the Sun) that may

have a similar composition.

On Earth, the Chapman cycle generates and maintains the ozone layer at an altitude of ∼25km, which is the primary atmospheric absorber of UV radiation. The Chapman cycle also interacts with other photochemical cycles, such as those of NOx and HOx species, further impacting the transmittance of UV radiation through the atmosphere. In addition to ozone, there are many other trace gases in the Earth's atmosphere that can undergo photolysis. At the surface, UV irradiation, has

implications for both prebiotic chemistry and extant life (Ranjan et al., 2017; Rimmer et al., 2018, 2021; Eager-Nash et al., 2024), and therefore, photochemistry is likely to play an important role in shaping the habitability of planets. In particular, for Earth-like exoplanets orbiting M-dwarfs, high levels of stellar activity can drive frequent and powerful emission of short-wavelength flux. Work exploring the cycling of ozone has been performed demonstrating elements, such as formation of

secondary ozone layers, and shielding from flaring caused by ozone build–up from previous flares (Chen et al., 2019; Yates





et al., 2020; Braam et al., 2022, 2024; Ridgway, 2023). Photolysis also likely played a key role in the formation of haze, potentially acting to shield the surface from UV radiation to some extent, during the Earth's Archean era, where life was first present (Arney et al., 2016; Mak et al., 2023; Eager-Nash et al., 2024).

In order to calculate photolysis rates, we need information on the absorption cross section of the species involved, the

quantum yield of the reactions, which is the branching ratio indicating which particular photolysis pathway is most probable, possible enhancement factors, due to photoionisation, and the spectrum of the star, particularly including the short wavelength flux. Cross sections and quantum yield data are measured in laboratory experiments, with their subsequent recommended values collated in various literature sources. Photolysis models have been used to perform detailed 1D intercomparison studies and provide benchmark photolysis rates, given the input data, in the context of Earth, such as that of CCMVal PhotoComp

2011 (Chipperfield et al., 2010, hereafter termed "PhotoComp").

The two-stream radiation scheme within Socrates, Suite-Of Community RAdiative Transfer codes based on Edwards and Slingo (1996), includes both a calculation of the radiative heating rates, and photolysis rates within a simulated atmosphere. The Socrates scheme is routinely used within the Met Office climate model, the Unified Model (UM), to simulate the climate and weather of Earth (Walters et al., 2019), as well as that of the Archean Earth (e.g Eager-Nash et al., 2023; Mak et al., 2023),

Mars (e.g McCulloch et al., 2023), terrestrial exoplanets (e.g. Mak et al., 2024) and a class of gaseous exoplanets termed 'hot Jupiters' (e.g Zamyatina et al., 2024). Socrates is also implemented in LFRic, the next-generation climate model of the Met Office (Adams et al., 2019), which is still in development, and has been coupled to other GCMs, such as ROCKE-3D and the University of Exeter's ISCA model. Previous incarnations of the UM have relied on the Fast-Jx scheme (Wild and Prather, 2000; Bian and Prather, 2002; Neu et al., 2007), as part of UKCA (Archibald et al., 2020), to treat photochemistry (see for example

Braam et al., 2022; Bednarz et al., 2019), where only wavelengths down to 177 nm are considered, as it is primarily for the study of the troposphere and stratosphere, where shorter wavelengths have been largely attenuated (Telford et al., 2013; Braam et al., 2022). The implementation within Socrates allows both extension of the model to higher parts of Earth's atmosphere, by including additional short-wavelength flux, and flexibility regarding the input stellar spectrum allowing application to planets and scenarios other than modern Earth. The inclusion of a photolysis scheme within Socrates was motivated by efforts to model

the effects of space weather, such as solar flares (Jackson et al., 2020). The treatment of photolysis needs to be considered within the general treatment of radiation transport, as it indirectly affects the heating rate of the atmosphere. Inclusion of the mesosphere and lower thermosphere also requires the inclusion of wavelengths down to far and extreme UV. In these parts of the atmosphere $O_2$, $N_2$ and O absorption occurs within the far and extreme UV wavelength ranges. The aim of this work is to benchmark the photolysis capabilities of this new scheme, for applications to both Earth and exoplanets.

The atmospheric compositions of terrestrial exoplanets are poorly constrained by current observations, so studies have focused on either adopting the atmospheric composition, sometimes simplified, of the modern Earth (e.g. Boutle et al., 2017; Cooke et al., 2023; Bhongade et al., 2024), or the Archean Earth (e.g. Eager-Nash et al., 2023; Mak et al., 2023) where the focus is on habitability. However, for the early-Earth and exoplanets, species in addition to those benchmarked in PhotoComp are required, such as $H_2O$, $CH_4$, $CO_2$ and many others.





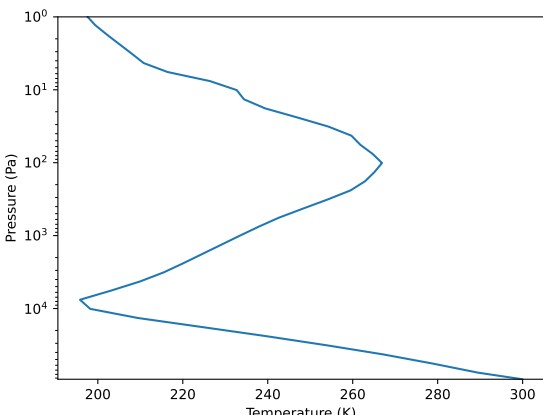

**Figure 1.** The pressure ( Pa)-temperature ( K) atmospheric profile from Chipperfield et al. (2010) adopted in this work.

In this work we benchmark Socrates photolysis rates at high resolution for species relevant to Earth and exoplanet atmo­spheres, validating against PhotoComp where possible, and extending to the study of new species and different stellar spectra. We collate up-to-date recommended cross-section and quantum yield sources, which were incorporated into the Socrates scheme, and extend on the low-resolution benchmarking previously preformed by Ridgway (2023) which only included the species $O_3$ and $O_2$. Specifically, we calculate photolysis rates for all our target species under Earth-like atmospheric structures,
and under irradiation from a solar or M dwarf spectrum.

The rest of this paper is structured as follows: Section 2 details the Socrates photolysis scheme. Next in Section 3, we sum­marise our data sources for the cross sections, the quantum yields and our solar spectrum. Section 4 presents our results and is split into two parts. The first part, Section 4.1, presents the rates calculated for Earth and compared with PhotoComp, cate­gorised by type, namely: Ox, HOx, NOx and organic. Then, using the same atmospheric profile but using Proxima Centauri's
stellar spectrum (the host star of a nearby, potentially 'Earth-like' exoplanet, Anglada-Escudé et al., 2016), we compare the rates yielded from this spectrum with those yielded by the solar spectrum, again separated into the categories used for Earth, in Section 4.2. Extra species relevant to exoplanets are included in an extra category for future reference. Finally, in Section 5 we provide our conclusions and indicate directions for future work.

## 2   Model Description

In this section, we detail the new Socrates photolysis scheme (Manners, 2024), describing how the rates are calculated alongside an overview of the radiative transfer calculation. Our specific configuration and setup are provided alongside the reasoning behind our choices.





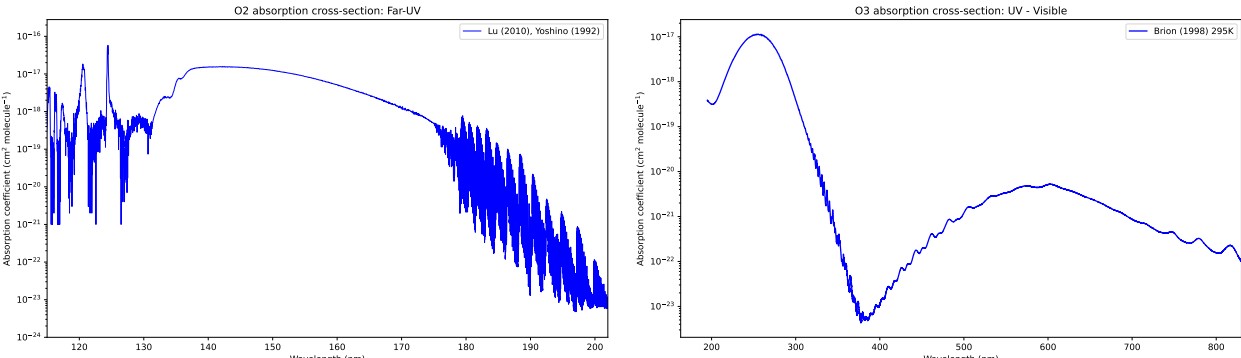

**Figure 2.** Absorption cross section ($cm^2$molecule$^{-1}$) against wavelength (nm) for the UV/visible range for $O_2$ & $O_3$ (*left* and *right panels*, respectively). The $O_2$ data are from Lu et al. (2010) and Yoshino et al. (1992), and that for $O_3$ from Brion et al. (1998) (see Table A1 for our full list of data sources).

## 2.1 Socrates Photolysis Scheme

The radiative transfer is calculated using the two-stream scheme within Socrates, solving for the radiative fluxes and heating
rates within the atmosphere using the absorption and scattering coefficients, and the input stellar/solar spectrum. A pseudo-spherical approximation is used whereby the plane-parallel approximation of the atmosphere is replaced by spherical shells (see Manners et al., 2024; Jackson et al., 2020; Christie et al., 2022, for details). This provides a more accurate calculation of the path for the direct beam and allows for illumination under twilight conditions.

The correlated-$k$ method is used for computational efficiency. Essentially, the wavelengths within a spectral band are re-
ordered in terms of increasing strength of absorption. Within a new cumulative probability space, as opposed to wavelength space, the wavelengths are binned so that similar coefficients are grouped together. Therefore, radiative flux calculations are performed for each absorption bin, or $k$ term.

However, for photolysis calculations a higher resolution is generally needed because within an interval the strength of absorption, actinic flux and quantum yield can all vary independently. To remedy this, the wavelength regions the $k$ term
represents, are designated as sub-bands, and 'remembered', so the calculated flux for each $k$ term can be mapped back to where it came from to form this new high resolution flux spectrum.

The actinic flux ($A$) is the integrated radiative intensity ($I$) over all directions ($\omega$), where $d\omega$ is the solid angle, and is needed to initiate the photochemical reaction. This is given by

$$A = \int_{4\pi} I d\omega. \tag{1}$$

A representative value of the actinic flux across a model layer is calculated from the two-stream fluxes using

$$A = \frac{-\Delta F}{\Delta \tau_{\mathrm{vert}}}, \tag{2}$$





where, $\Delta F$ is the total flux divergence and $\tau_{\text{vert}}$ is the vertical optical depth to absorption. The actinic flux is calculated per $k$ term in units of $W m^2$ ($F_A$) and then it is mapped back to the sub-bands. The flux in terms of number of photons is required for the calculation of the photolysis rates which is why at this point, it is converted into units of photons $m^2 \ s^{-1}$ ($A$) by dividing by the energy of the corresponding photon at the mid-point wavenumber for the sub-band. Given that the sub bands are very narrow, the central frequency is used, as opposed to using the flux distribution within the band to determine where the photon energy originates from.

The photolysis rate, $J$ with units $s^{-1}$, is calculated using,

$$J = \int \sigma Q A d\lambda, \tag{3}$$

where $\sigma$ is the absorption cross section of the molecule, $Q$ is the quantum yield or branching ratio which is the number of molecules undergoing a photochemical event per absorbed photon, and $A$ is the actinic flux. In Socrates the equation takes the form

$$J = \frac{m}{N_A hc} \sum_{\text{k-terms}} F_A \sum_{\text{sub-bands}} k_{\text{abs}} Q \lambda w, \tag{4}$$

where $m$ is the molecular weight of the absorbing species, $N_A$ is Avogadro's number (mol$^{-1}$), $h$ is Planck's constant (J s), $c$ is the speed of light, $F_A$ is the actinic flux in W m$^{-2}$, $k_{\text{abs}}$ is the mass absorption coefficient (m$^2$ kg$^{-1}$) of the species undergoing photolysis, $Q$ is the quantum yield, $\lambda$ is wavelength and $w$ is the fraction of the the actinic flux in the sub-band, or the sub band weight. The proportion of the flux divergence used for photolysis can be immediately released for atmospheric heating or can be removed from the radiative heating rates diagnosed by the scheme. This allows for later exothermic release of the absorbed energy by an external chemistry scheme. More details and descriptions of these processes can be found in Manners (2024).

This photolysis scheme has the capabilities to account for temperature and pressure dependencies of the cross sections and temperature dependencies of the quantum yields, which is needed for Earth applications as well as exoplanets. The photolysis scheme in Socrates is not intrinsically tied to the Solar spectrum thereby allowing different input spectra, and the fraction of flux within the sub-band can alter accordingly. For Socrates, a configuration file, known as a 'spectral file', contains all the relevant information allowing for calculation of the radiative fluxes and therefore heating rates, as well as the photolysis rates. These files contain information on spectral band wavelength ranges, gaseous absorption coefficients ($k$ terms), aerosol/cloud properties, photolysis pathways and their quantum yields, and the stellar spectrum. In the following section we detail the spectral file configuration constructed for this work.

### 2.1.1 2000 Band Configuration

For this study we have constructed a high resolution 2000 band spectral file. The first 1000 bands are 1 nm wide (0.9 nm for the first band) and cover the wavelength range 0.1-1000 nm. The number of sub bands over this range is 13799, providing the resolution used for photolysis which can be seen in the spectral plots in section 4. Bands 1001-2000 have a resolution of 10 cm$^{-1}$ and cover the range 1000 nm-0.01 m. This wide range allows for complete coverage of stellar and thermal radiative transfer, allowing for calculations across a range of atmospheres. However, for the calculation of the photolysis rates, only





wavelengths less than 1100 nm were considered. The absorption coefficients ($k$ terms) are calculated from the relevant input
cross sections. These cross sections are taken from the sources described in Section 3, and listed in Table A1, alongside the
photolysis pathways and branching ratios we adopted. The number of $k$-terms varies up to a maximum of 22 per band for the
major gases.

In this study we include species that are important for Earth's stratosphere and were also part of PhotoComp. These species
fall broadly into the categories: Ox, NOx and HOx, and organic species relevant to Earth. These selected species were chosen to
compare with the output of the radiative transfer code Fast-Jx (Wild and Prather, 2000; Bian and Prather, 2002; Neu et al., 2007)
used for the Regional Air Quality (RAQ) mechanism (Savage et al., 2013; Mynard et al., 2023). For species, such as $HNO_4$, a
small portion of photolysis is initiated by radiation at near infrared wavelengths. For the purpose of the intercomparison and
as there are no sources of opacity in the infrared in our calculations, this was neglected. The species in addition to those within
PhotoComp that we have added for exoplanets align with the high temperature network of Venot et al. (2012) which is designed
for hot hydrogen-dominated exoplanets, such as hot Jupiters as this is intended for future studies of these objects. There are
hydrocarbon species, such as methane, $CH_4$, as well as water, that also have relevance for potential studies of early-Earth like
conditions on exoplanets.

We incorporate Rayleigh scattering from $O_2$ and $N_2$ (air) down to 175 nm. This limit coincides with the threshold for $O_2$
$\rightarrow O(^3P) + O(^1D)$ photolysis. It is assumed that most of the absorbed flux from shorter wavelengths is used for dissociation.
Therefore, Rayleigh scattering by air is not suitable and can be neglected below 175 nm.

## 3 Input Data

The three most prominent data sources for absorption cross sections and quantum yields are the recommendation from the JPL
report (Burkholder et al., 2020), the IUPAC recommendations (Atkinson et al., 2004) and others collated within sources such
Venot et al. (2012) and Hébrard (priv. comm. 2022). Many of the recommended sources were retrieved from the MPI-UV/Vis
database (Keller-Rudek et al., 2013). Though, for some species high resolution cross section or line list data was needed and
sourced elsewhere. A full list of our data sources is presented in the Appendix A as Table A1. Photoelectron enhancement
factors are also included for some reactions where the data were readily available. Photoionisation can free electrons and
induce more reactions and the quantum yield must adjust accordingly via these factors. This process was only included for
oxygen as the data was available and predominantly comes into effect for EUV wavelengths (Solomon and Qian, 2005).

For the species relevant to exoplanets, much of the data was taken from Venot et al. (2012). Other exoplanet species data
coincided with what we used for the Earth species and are detailed in the Table A1.

The solar spectrum we use for this work is the CMIP6 recommendation from Matthes et al. (2017) averaged over solar
cycle 23 from September 1996 to December 2008. For wavelengths shorter than 10 nm, the spectrum used by Solomon and
Qian (2005) was included. The spectrum we use for Proxima Centauri is the same used in the work of Ridgway (2023). This
spectrum is a combination of two sources: the MUSCLES survey (France et al., 2016; Youngblood et al., 2016; Loyd et al.,
2016) and Ribas et al. (2017).





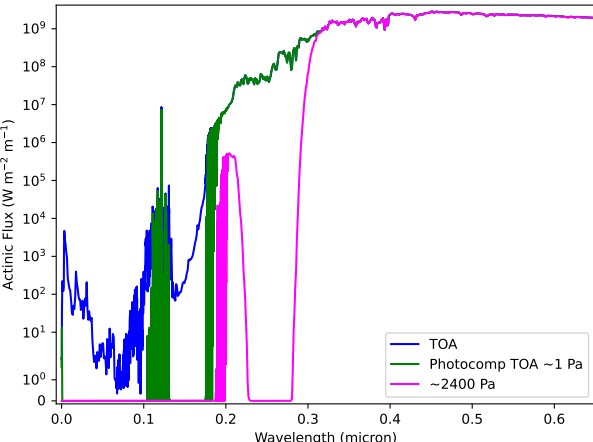

**Figure 3.** Actinic flux ($W\,m^{-2}\,m^{-1}$) as a function of wavelength ($\mu$ m) at three different levels, the top-of-atmosphere, upper mid-atmosphere (a pressure of 1 Pa) and lower mid-atmosphere (at a pressure of $\sim$2400 Pa) corresponding to the ozone layer, shown by the solid blue, green, and magenta lines, respectively.

## 4 Results: Testing the Scheme

In this section we first present a comparison between our calculations and those presented in PhotoComp. Firstly, we outline the setup (Section 4.1.1) and present our calculated solar actinic flux (Section 4.1.2), before presenting rates for our different categories of species; namely Ox, HOx, NOx and Organic. We then move to an input spectrum of Proxima Centauri (Section 4.2), again presenting the calculated actinic flux (Section 4.2.1) and comparing the rates against those calculated for a solar spectrum grouping species by the same categories as used for Earth, but including an additional section for those species added for later applications to exoplanets (Section 4.2.6).

### 4.1 Benchmarking: PhotoComp

#### 4.1.1 Setup

The Chemistry-Climate Model Validation, Stratosphere- troposphere Processes And their Role in Climate Evaluation (SPARC) CCMVal-2, was a climate model intercomparison initiative, which included an element on the benchmarking of photolysis models, PhotoComp 2008. The results were produced in the subsequent report, Chipperfield et al. (2010). Initially conducted in 2008 using JPL 2006 data (Sander et al., 2006), PhotoComp was repeated in 2011 (Chipperfield et al., 2013) with predominantly JPL 2010 data (Burkholder et al., 2020). The primary goal of this photolysis intercomparison was to evaluate how different models calculated the photolysis rates in the stratosphere and troposphere. Part 1a of their experimental set-up was used as a basis for our own. There they assumed a clear sky, no aerosols, high sun (Solar Zenith Angle, SZA = 15°) over the ocean,



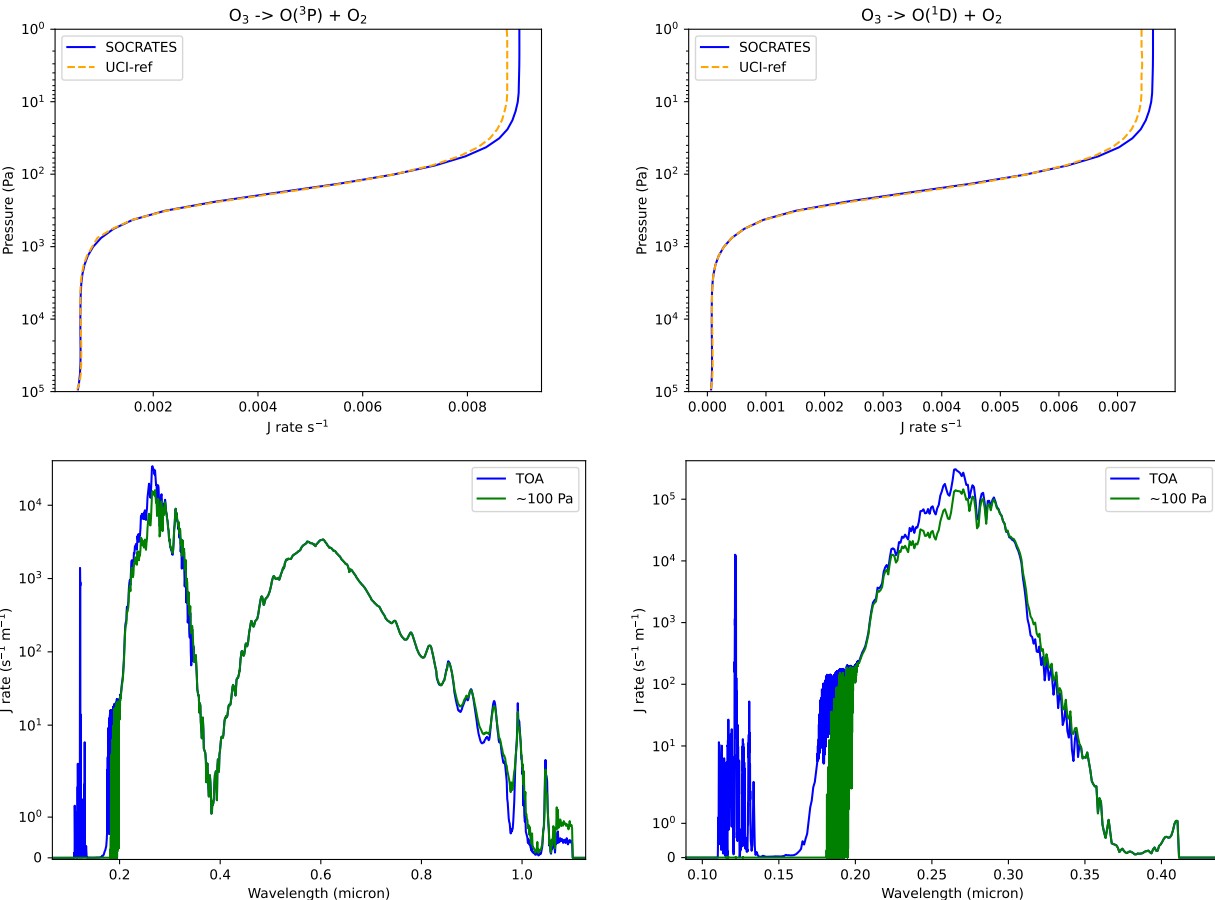

**Figure 4.** Photolysis rates (J rate s$^{-1}$) as a function of pressure (Pa, *top* row), and as a function of wavelength ($\mu$ m, *bottom row*), for two dissociations of O$_3$. Namely, dissociation to O$_2$ and the ground state of atomic oxygen, O($^3$P), and dissociation of O$_3$ into oxygen, O$_2$, and the excited state of atomic oxygen, O($^1$D) (*left* and *right columns*, respectively). The rates from Chipperfield et al. (2010) and this work are shown as the dashed orange, and solid blue lines, respectively (*top* row), and those at the TOA and ~100 Pa as the blue and green lines, respectively (*bottom row*).




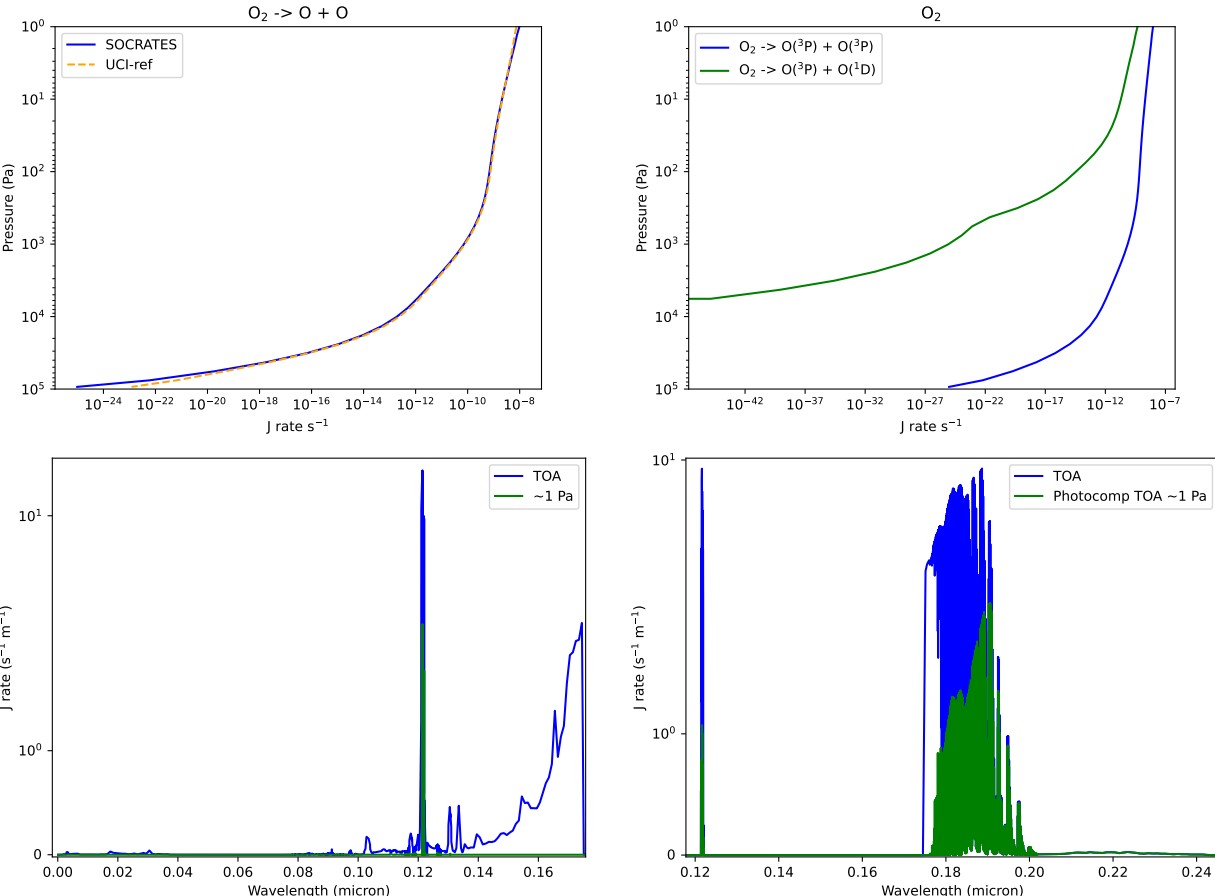

**Figure 5.** Total photolysis rates of $O_2$ into two atomic O, as well as the separate rates for the dissociations $O_2 \rightarrow O(^3P) + O(^1D)$ and $O_2 \rightarrow O(^3P) + O(^3P)$ as a function of pressure (Pa) (solid green and blue line respectively), shown in the *top* panel. The *bottom* panel shows the rates for these dissociations $O_2 \rightarrow O(^3P) + O(^1D)$ *left* and $O_2 \rightarrow O(^3P) + O(^3P)$ *right* as a function of wavelength ($\mu$ m) at the TOA and for a pressure of $\sim 1$ Pa which corresponds to PhotoComp's TOA (solid blue and green lines, respectively).





(albedo = 0.10, Lambertian), the incoming solar irradiance at top-of-atmosphere is taken as 1365 W m$^2$ and Rayleigh scattering is included. The pressure-temperature profile used by PhotoComp, and adopted here is shown in Figure 1.

The two reference models from PhotoComp we compare with[1] are: the UCI reference model (hereafter, UCI-ref, Prather, 1974; Wild and Prather, 2000; Bian and Prather, 2002) and Fast-Jx as implemented and run by UCI (hereafter, UCI-Jxr, Prather, 1974; Wild and Prather, 2000; Bian and Prather, 2002). The UCI-ref model is a photochemical 1D box model that implements 77 wavelength bins and 3-6 sub-bins. The UCI-Jxr utilises 18 wavelengths bins and uses version 6.2 of Fast-Jx, another 1D photochemical model. Both reference models are valid to an altitude of ∼64 km or ∼10 Pa [2]. The solar spectrum used in

the UCI reference models is the Solar Ultraviolet Spectral Irradiance Monitor (SUSIM) spectrum and is an average of two high and low points within the solar cycle which occurred on 29 March 1992 and 11 November 1994. For our comparison of the photolysis rates (Sections 4.1.3, 4.1.4, 4.1.5 and 4.1.6) in most cases the UCI-ref and UCI-Jxr results are indiscernible, therefore we only show the former, but present both models in cases where they differ. For our calculations, we adopt the O$_3$ abundance used in PhotoComp, and an O$_2$ abundance of Earth's atmosphere as sourced from Anderson et al. (1986).

The PhotoComp study also included tests of the accuracy of the actinic flux calculations for different atmospheric compositions. However, the accuracy of the Socrates radiative transfer calculations has been extensively validated previously for both Earth (Pincus et al., 2020) and exoplanets (Amundsen et al., 2014), therefore we restrict our work here to benchmarking the photolysis rates only. The region of interest for these photolysis rates is primarily the stratosphere extending into the mesosphere. At this point in the atmosphere, ozone and oxygen are the main absorbers in the UV/visible range. Therefore, it is their

abundance that is the main determinant of the actinic flux available for all the species undergoing photolysis.

    PhotoComp calculations extend to a shortest wavelength of 177.4 nm, thereby omitting Lyman-$\alpha$ absorption. However, for our results we use cross section data that includes shorter wavelengths in the extreme UV (EUV) range, which also requires inclusion of N$_2$, O and N abundances (Anderson et al., 1986) as these are the main absorbers at EUV wavelengths, shortward of 100 nm. Additionally, the photolysis of NO and its absorption are affected by the interplay with O$_2$ absorption in the Schumann

Runge bands. Therefore, we include an abundance of NO in our calculations using a value for Earth's atmosphere (Anderson et al., 1986), for the calculation of the relevant photolysis rates. We include two additional atmospheric layers (taking the total to 42) at the top of the model domain containing O, N [3] and N$_2$ (Anderson et al., 1986) in order to account for the attenuation of shorter (EUV) wavelengths that occurs at high altitudes. However, we only present results up to a model level of 40 for comparisons to PhotoComp throughout this work. To test the impact of these additional layers and wavelengths, we performed

calculations where fluxes at wavelengths shorter than 177 nm were omitted from the calculations as well as only including O$_3$ and O$_2$, to better match the PhotoComp setup. This revealed a negligible impact on the results for most species, but is noted where relevant. Finally, we only present results for the main species in the main part of this paper. The main species are those pertinent to Earth's key photochemical cycles (Chapman, HOx and NOx) as well as some organic species. Note that where appropriate ($^3$P) refers to the ground state of the atom and O($^1$D) being an excited state.

---

[1]data provided by Martyn Chipperfield and retrieved from https://homepages.see.leeds.ac.uk/ lecmc/sparcj

[2]retrieved from the accompanying notes/directories from https://homepages.see.leeds.ac.uk/ lecmc/sparcj Chipperfield et al. (2010)

[3]Taken from the Community Coordinated Modeling Center VITMO ModelWeb Browser Results, MSISE-90 model listing database





### 4.1.2 Actinic Flux

As photolysis is driven by short-wavelength flux, species and bands that absorb UV and visible light have a direct impact on the resulting photolysis rates as they dictate the actinic flux. The gases $O_2$ and $O_3$ are the main absorbers in this regime, and their absorption cross sections are shown in Figure 2. The major bands for $O_3$ are the Hartley bands 200-310 nm, which predominantly absorb in the stratosphere, with additional absorption longward of 310 nm through, for example, the Huggins and Chappuis bands. For $O_2$, absorption is mainly via the Schumann Runge bands (175 to 205 nm) and continuum (130 to 175 nm), as well as absorption of solar Lyman-$\alpha$ emission (121.45 to 121.7 nm).

Figure 3 shows the actinic flux, as calculated by Socrates using Equation 2 and detailed in Section 2.1, at the top-of-atmosphere (TOA, solid blue line), upper mid-atmosphere (at a pressure of 1 Pa) which represents the top model level specified by PhotoComp as used by the UCI-ref model (green line), and the lower mid-atmosphere (at a pressure of $\sim$2400 Pa) corresponding to the ozone layer (magenta line). The absence of actinic flux between 220-290 nm, for the lower/mid-atmosphere is due to ozone absorption within the Hartley bands.

### 4.1.3 Ox

Figure 4 shows the rates for two possible dissociation reactions for ozone, namely $O_3 \rightarrow O(^3P) + O_2$, and $O_3 \rightarrow O(^1D) + O_2$ as the *left* and *right* columns, respectively, and as functions of pressure and wavelength as the *top* and *bottom* rows, respectively. The *top* row of Figure 4 shows the rates from both the UCI-ref (dashed orange line) and this work (solid blue line), demonstrating excellent agreement apart from a slight difference towards the top of the atmosphere caused by differences in the input data (cross sections, quantum yields and stellar spectrum) between the models. The *bottom* row of Figure 4 shows the rates at the TOA and $\sim$100 Pa which approximately corresponds to the point where the models begin to slightly disagree. The Socrates rates were also recalculated without contributions from wavelengths shorter than 177 nm. This had a negligible impact on the results, indicating that the inclusion of Lyman-$\alpha$ emission is not significant.

Figure 5 shows the total dissociation rate for $O_2 \rightarrow O + O$ for both the UCI-ref (orange dashed line) and our work (solid blue line) as a function of pressure ( Pa) in the *top left* panel, as well as the separate Socrates rates for the dissociations $O_2 \rightarrow O(^3P) + O(^1D)$ and $O_2 \rightarrow O(^3P) + O(^3P)$ (solid green and blue line respectively) in the *top right* panel. In the *bottom* panels the rates are shown as a function of wavelength for the separate reactions. The *top left* panel of Figure 5 again shows excellent agreement between the rates calculated using Socrates and the UCI-ref values, with only a slight departure at very low pressures ($\sim$10 Pa, or above $\sim$64 km) where the UCI-ref model is no longer valid as it does not include EUV wavelengths (see discussion in Section 4.1.1). The *top right* panel shows that the $O_2 \rightarrow O(^3P) + O(^3P)$ reaction is the main contributor to the total dissociation rate, while the $O_2 \rightarrow O(^3P) + O(^1D)$ reaction only contributes for wavelengths below the threshold at 175nm (spectrum, *bottom left* panel). Repeating this comparison while omitting flux at wavelengths $\leq$177 nm reduces the disparity between the Socrates and UCI-ref results to negligible levels (not shown).




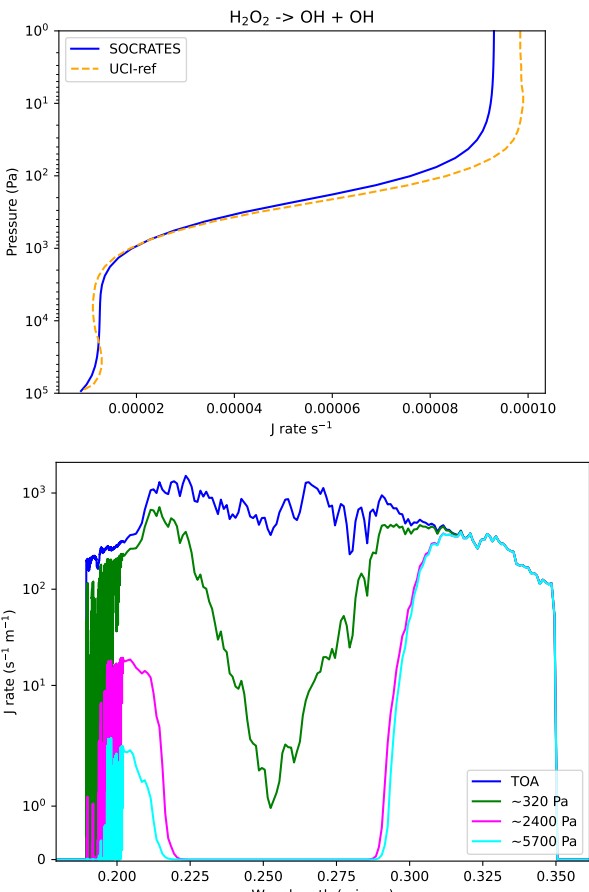

**Figure 6.** Photolysis rates of $H_2O_2$ into two OH, as a function of pressure (Pa, *top* panel), and as a function of wavelength ($\mu$ m, *bottom* panel). For the rates as a function of wavelength (*bottom*) values are show at the TOA and for pressures of $\sim$320, $\sim$2400 (which approximately coincides with the ozone layer) and $\sim$5700 Pa (solid blue, green, magenta and cyan blue, lines, respectively).

### 4.1.4 HOx

Figure 6 shows the photolysis rates for the reaction $H_2O_2 \rightarrow OH+OH$ as a function of pressure (Pa) in the *top* panel and as a function of wavelength ($\mu$ m) in the *bottom* panel. A distinct drop in the photolysis rates for wavelengths of 220-290 nm is present in the mid-atmosphere. This is caused by the drop in actinic flux, as shown in Figure 3, due to flux absorption from the

Hartley bands for ozone. Again the match of our calculated rates with those of UCI-ref are good, although slight departures are present towards lower pressures and between $\sim 10^3$ and $\sim 10^4$ Pa. These small discrepancies can be attributed to differences in input data as discussed in the ozone case in Section 4.1.3.



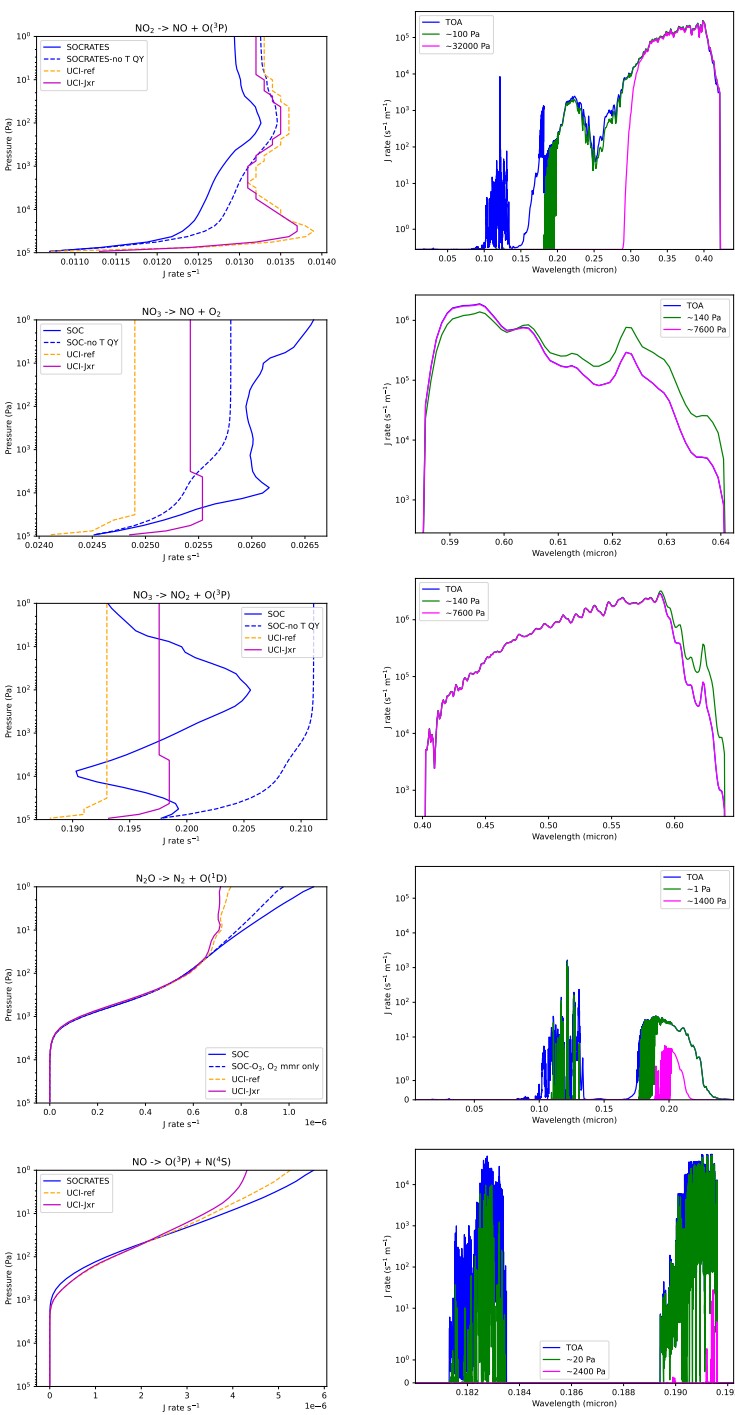

**Figure 7.** Photolysis rates of the reactions for the NOx species: as a function of pressure (Pa, *left* panel) with the added UCI-Jxr (purple line), and as a function of wavelength ($\mu$ m, *right* panel).




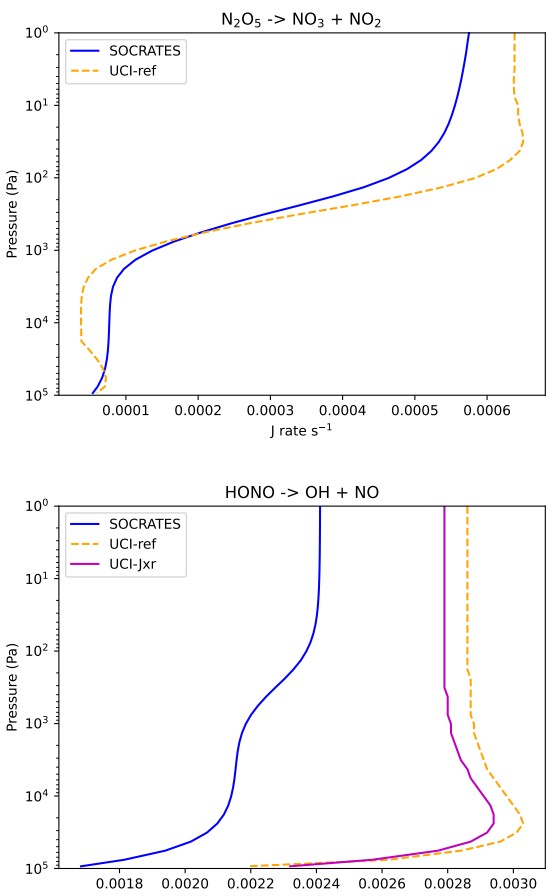

**Figure 8.** Photolysis rates yielded from the reaction $N_2O_5 \rightarrow NO_3 + NO_2$ and $HONO \rightarrow OH + NO$ against atmospheric pressure( Pa).

### 4.1.5 NOx

Figure 7 shows the photolysis rates for the reactions $NO_2 \rightarrow NO + O(^3P)$, $NO_3 \rightarrow NO + O_2$, $NO_3 \rightarrow NO_2 + O(^3P)$, $N_2O \rightarrow N_2 + O(^1D)$ and $NO \rightarrow N(^4S) + O(^3P)$ as a function of pressure ( Pa, *left* panels) with UCI-Jxr added (purple line) and as a function of wavelength ($\mu$ m) in the *right* panels.

With regard to the reaction $NO_2 \rightarrow NO + O(^3P)$, the rates from Socrates and UCI-ref in Figure 7 generally match in terms of shape, due to the temperature dependent absorption cross sections. However, when a temperature dependent quantum yield is introduced, our results are displaced slightly to the left, thereby reducing the overall photolysis rates.

The absorption spectrum for $NO_2$ is quite complex in the near-UV and visible (Akimoto, 2016). High-resolution absorption data was used for this reason. Figure 7 shows $NO_2$ photolysis leading to the production of NO continues out to around 420 nm. This is beyond the threshold limit for this photolysis reaction to occur which is 398 nm, but can be accounted for by additional




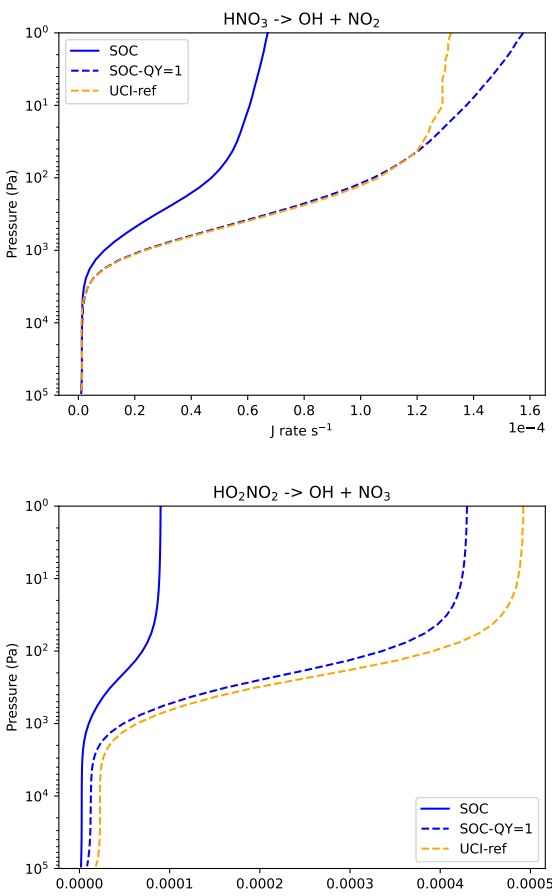

**Figure 9.** Dissociation of $HNO_3$ into OH and $NO_2$ (*top* panel) and dissociation of $HO_2NO_2$ into OH and $NO_3$ (*bottom*) against atmospheric pressure( Pa). The photolysis rates yielded by Socrates match more closely to the reference (orange) when the quantum yield is 1.

internal and translational energy supplied to the molecule (Akimoto, 2016; Burkholder et al., 2020). This reflects the behaviour of the quantum yield which decreases down to zero between 300-420 nm.

The peak in photolysis at $\sim$100 Pa as shown in Figure 7 (*top left*) is due to the temperature dependent cross-sections with a further exaggeration of the peak seen when T dependent quantum yields are used in Socrates. While the peak at $\sim$100 Pa is similarly represented in Socrates and the reference models, the peak seen in the reference models at $\sim$32000 Pa is not seen in Socrates. Figure 3 shows that the actinic flux appears to be more attenuated over the critical wavelengths < 290 nm, which is likely due to updated ozone absorption cross-sections for HITRAN 2020 (see Table A1).

For the reaction $NO_3 \rightarrow NO + O_2$, the second row of Figure 7, the reference and Socrates calculated rates match approximately well in terms of shape and order of magnitude without the inclusion of temperature dependent quantum yields. However, with the inclusion of this temperature dependence, the results differ more significantly. For the reaction $NO_3 \rightarrow NO_2 + O(^3P)$,





the third row of Figure 7 shows the opposite is true. Although the change of the rates with pressure has changed drastically, the rates are closer to the reference models after the inclusion of the temperature dependent quantum yield.

The *right* panels of Figure 7 showing the photolysis spectra for $NO_3$ shows that the rates as a function of wavelength are similar for the mid-atmosphere and the TOA, generally diverging more for longer wavelengths. A high-resolution absorption cross section was used as $NO_3$ has very strong absorption bands with vibrational structures affecting the wavelengths $400-700\,nm$, especially around 600-700 nm. However, the photolysis spectra show a sharp cutoff point at around 640 nm, where the rates drop to zero, caused by the quantum yield $\phi$ ($O + NO_2$) and $\phi$ ($NO + O_2$) falling to zero for wavelengths above $\sim640\,nm$. It

is also important to note that between 400-640 nm that there are no significant sources of absorption of flux. The difference in the rates as a function of pressure between our calculations and that of the reference, mimics the pressure-temperature profile, Figure 1, and is almost solely due to the temperature dependent quantum yield.

The TOA Socrates rates for dissociation $N_2O \rightarrow N_2 + O(^1D)$ are higher than the reference however, similarly to the oxygen species, we include EUV wavelengths in the cross sections. Similar to the $O_2$ case as discussed in Section 4.1.3, high TOA rates

at these shorter wavelengths are displayed in the photolysis spectrum as shown in Figure 7. With only $O_3$ and $O_2$ absorption considered, and a cut-off at 177 nm, the rates near the TOA are higher but reduced and more closely match the reference, indicating that discrepancies lie with different input cross sections and model differences.

The gas NO's photoabsorption cross section features fine band structures. Fluorescence occurs, except for in the bands $\delta$(0-0) and $\delta$(1-0), which correspond to the wavelengths $189.4-191.6\,nm$ and $181.3-183.5\,nm$ respectively (Akimoto, 2016; Mayor

et al., 2007). It is in these regions that photolysis occurs with a quantum yield of unity. The photolysis spectra for the NO dissociation clearly shows these regions through the presence of high photolysis rates. These regions coincide with the $O_2$ Schumann Runge bands. For these reasons an accurate rotational line list is needed and this was sourced from the line list, XABC, from Exomol (Tennyson et al., 2016; Wong et al., 2017). Therefore a pressure and temperature dependence, of the absorption coefficients, were included in our calculations. The rates calculated by Socrates match the reference values well, as

shown in Figure 7, indicating that similar line lists were also likely in the reference. Note that an NO mass mixing ratio was included in the calculation of the photolysis rates as discussed in Section 4.1.1.

Figure 8 shows the photolysis rates yielded from the reaction $N_2O_5 \rightarrow NO_3 + NO_2$ and $HONO \rightarrow OH + NO$ against atmospheric pressure ( Pa). Briefly, for $N_2O_5$ the Socrates and reference rates match well with differences accounted for by differences in input data similar to the ozone case 4.1.3. For HONO the rates are offset to lower values. This could indicate a missing

component either in the wavelength range included or the quantum yield, or the use of different input data.

Species $HNO_3$ and $HO_2NO_2$ also show discrepancies between the Socrates and reference rates, shown as a function of pressure in Figure 9. For $HNO_3$, we adopt the quantum yield given by IUPAC, but only obtain a match to the reference rates when we assume a quantum yield of one (as also found by other studies, e.g. Burkholder et al., 2020; Ridgway, 2023). Finally, $HO_2NO_2$ also shows a significant discrepancy between the Socrates and reference rates, unless we adopt a quantum yield of

one and the near–infrared cross sections and quantum yields are omitted.




### 4.1.6 Organic

Figure 10 shows the photolysis rates for the reactions $H_2CO \rightarrow H + HCO$, $H_2CO \rightarrow H_2 + CO$ and $OCS \rightarrow CO + S(^3P)$ as a function of pressure (Pa), as the *top*, *middle* and *bottom* panels, respectively. For Formaldehyde, $H_2CO$, our rates increase in relation to the reference calculations towards altitudes above the ozone layer, $\sim$25 km. As Formaldehyde features fine structure within its cross section, as a function of wavelength ($\mu$m), we have adopted high resolution input data which the reference models may not. Similarly to $H_2O_2$ (Figure 6 in Section 4.1.4), the photolysis rates for $H_2CO$ dissociation are impacted by the ozone layer in the mid atmosphere for wavelengths between 290-360 nm. As before, flux at shorter wavelengths has been included in the Socrates calculations, which is omitted in the calculation of the reference rates, but this does not contribute to the overall TOA photolysis rate similar to the ozone case detailed in Section 4.1.3. The main difference in our rates and those of the reference is most likely due to differences in choice of input data. Note that we implemented a quantum yield for standard pressure (1 atmosphere) and 300 K as recommended by JPL 19-5 (Burkholder et al., 2020) and that pressure and temperature dependence have not been taken into account.

Figure 10 shows that for Carbonyl sulphide, OCS, the Socrates rates and those of the reference agree very well. The species OCS has a strong temperature dependence of the cross sections, included in our calculations, with the agreement suggesting that the reference also included this.

Figure 11 displays the photolysis rates for the remaining organic species considered, namely $CH_3OOH$, $CH_3CHO$, PAN, $CH_3COCH_3$, CHOCHO, $CH_3ONO_2$, $CH_3COCHO$ and $HOCH_2CHO$, as a function of pressure ( Pa). For the first reaction pathway of polyacrylonitrile $CH_3C(O)OONO_2$, or PAN, the Socrates rates match the reference particularly well, whereas for the second PAN reaction and $CH_3COCH_3$ there is a significant discrepancy. The quantum yields adopted for our calculations of the rates for PAN and $CH_3COCH_3$ (based on the JPL recommendations) could be the source of this discrepancy. The quantum yields are temperature and pressure dependent for $CH_3COCH_3$, however we currently only have the functionality to represent the temperature dependence. To account for the pressure dependence we used an appropriate tropospheric pressure in the formulation to calculate the quantum yields for the four temperatures in the look-up table (see table A1). Future work is needed to properly incorporate the pressure dependence and this could be a contributing source of the difference with the reference model for $CH_3COCH_3$ as well as some other organic species (e.g. formaldehyde). There are even more significant differences between our rates and those of the reference for other organic species, namely, CHOCHO (Glyoxal), $CH_3ONO_2$ (methyl nitrate) and $CH_3COCHO$ (methylglyoxal). For the first reaction of CHOCHO our Socrates rates are higher than those of the reference, while for the second they are lower. The rates calculated by Socrates $CH_3ONO_2$ are also significantly higher than the references. The rates for $CH_3COCHO$ are lower and the plots display marked differences in shape between the profiles. For all species there is a great deal of uncertainty regarding the recommended input data as comparative studies are sparse.

### 4.2 Comparison with M Dwarf Spectra

Many M dwarf stars have been shown to host potentially Earth-like planets (Tuomi et al., 2019). For such planets, photolysis is likely to play an important role in determining the climate. Previous studies have explored the impact of both the quiescent





stellar irradiation and the impact of flares on the atmospheres of planets orbiting M dwarfs (e.g Ridgway, 2023). However, such
studies have focused on limited photochemical reactions and have not been extensively benchmarked. In this work, we perform
calculations using Socrates, with the irradiation of an M dwarf, to provide a set of initial benchmark rates for the major species
and photolysis reactions.

Our nearest star, Proxima Centauri (hereafter "Prox Cen") has also been shown to host a potentially Earth-like exoplanet
(Anglada-Escudé et al., 2016). Therefore, we adopt the spectrum of Prox Cen from Ridgway (2023), which is a combination
of data from the MUSCLES survey (France et al., 2016; Youngblood et al., 2016; Loyd et al., 2016) and Ribas et al. (2017).
However, we maintain the same total TOA incoming flux at $1365\,\mathrm{W\,m^2}$ as used for the Solar calculations in Section 4.1 to
make comparison between the resulting rates easier. This is an appropriate total incoming flux for a planet in the habitable zone
around Prox Cen. The Prox Cen and Solar spectra used are shown in Figure 12. As noted by Ridgway (2023) the spectrum of
Prox Cen has a higher proportion of FUV to X-ray flux than the Solar spectrum, particularly below $\sim$125 nm with the Prox Cen
flux for Lyman-$\alpha$ emission $\sim$ 121.6nm being significantly higher than the Solar flux. This has implications for the photolysis
rates of certain species, where the threshold wavelengths of the photolysis reactions are close to this point.

### 4.2.1   Actinic Flux

The higher levels of FUV and EUV flux for Prox Cen, compared to the Solar spectrum, shown in Figure 12, result in a greater
availability of actinic flux to drive photolysis below $\sim$175 nm. Figure 13 shows the actinic flux at three different atmospheric
pressure levels, namely the TOA (blue), $\sim$20 Pa (upper mid-atmosphere, green) and $\sim$2400 Pa (lower mid-atmosphere, corre-
sponding to the location of the ozone layer, magenta line) for both the Solar and Prox Cen spectra as the *top* and *bottom* panels,
respectively. Figure 13 shows that below 175 nm, there is about an order of magnitude higher actinic flux for the Prox Cen
spectrum compared to the Solar case, whilst at wavelengths greater than 175 nm there is significantly more actinic flux from
the Solar spectrum.

### 4.2.2   Ox

Figure 14 shows the photolysis rates for dissociations of $O_3 \rightarrow O_2 + O(^3P)$ and $O_3 \rightarrow O_2 + O(^1D)$ on a log scale as a function of
pressure (Pa, *top* panels) and wavelength ($\mu$m, *bottom* panels) for just the Prox Cen spectrum. The Prox Cen photolysis rates for
$O_3 \rightarrow O_2 + O(^3P)$ are much lower than Solar because the visible part contributes a larger proportion of the total rate and there
is much less NUV when compared with the Solar spectrum results (see figure 4). As a result, the sharp decline in photolysis
across the ozone layer is not seen as with the Solar case. There is also a larger contribution from Lyman-$\alpha$ wavelengths leading
to the rise in rates towards top-of-atmosphere for Prox Cen.

Figure 15 shows the dissociation rates of $O_2$ into $O(^3P)$ and $O(^1D)$ and $O_2$ into $O(^3P)$ and $O(^3P)$. For the rates as a function
of pressure, *top* panel, the Prox Cen values (red) are compared to that of the Solar case (blue). The Prox Cen rates for $O_2 \rightarrow$
$O(^3P) + O(^1D)$ are much higher than Solar (by about an order of magnitude) in agreement with Ridgway (2023). For the rates
as a function of wavelength ($\mu$m), *bottom* panel, the results are shown for the Prox Cen calculation at the TOA (blue) and a
pressure of $\sim$1 Pa (green), showing a much larger contribution from Lyman-$\alpha$ emission compared to the Solar case (compare





to Figure 5). For $O_2 \rightarrow O(^3P) + O(^1D)$ all contributions to the rates originate from wavelengths $< 175$nm where the Prox Cen actinic flux is greater, whereas for $O_2 \rightarrow O(^3P) + O(^3P)$ the major contribution is from wavelengths $> 175$nm leading to significantly higher rates for the Solar case.

### 4.2.3 HOx

As an example of photolysis of HOx species, we focus on the dissociations of $H_2O$, and specifically the reactions: $H_2O +$ $h\nu \rightarrow OH(X^2\Pi) + H$ where $OH(X^2\Pi)$ is a ground state, $H_2O + h\nu \rightarrow O(^1D) + H_2$, and $H_2O + h\nu \rightarrow O(^3P) + H_2$. Figure 16 shows the rates for these dissociations as yielded by the Solar and Prox Cen spectra (blue and red lines respectively) against pressure on a log scale *left* column, and as a function of wavelength ($\mu$ m) for the Solar case *middle* column and Prox Cen case *right* column at the TOA (blue) and $\sim20$ Pa (green). The dissociation $H_2O_2 \rightarrow OH + OH$ is also displayed in Figure 16 for reference.

Towards TOA the Prox Cen rates for all the $H_2O$ reactions are an order of magnitude higher than those of the Solar case due to the strong contribution from the Lyman-$\alpha$ region. High up in the atmosphere, as indicated in Figure 13, we see that the TOA actinic fluxes are quickly attenuated below $\sim180$nm due to $O_2$ absorption in the Schumann-Runge bands and continuum leaving only significant contributions from the Lyman-$\alpha$ emission line and, for the case of $H_2O + h\nu \rightarrow OH(X^2\Pi) + H$, wavelengths greater than 180nm by $\sim20$ Pa. Figure 16 shows that the production of $H_2$ from $H_2O$ photolysis occurs predominantly in the wavelength region $\sim0.08$-$130$ nm and so similar to the oxygen case as detailed in Section 4.2.2, higher rates are seen for Prox Cen at all levels of the atmosphere caused by the increased actinic flux.

### 4.2.4 NOx

For photolysis of the NOx species, Figure 17 shows the rates for the dissociation of $NO_2 \rightarrow NO + O(^3P)$, $NO_3 \rightarrow NO + O_2$, $NO_3 \rightarrow NO_2 + O(^3P)$, $N_2O \rightarrow N_2 + O(^1D)$ and $NO \rightarrow N(^4S) + O(^3P)$ as yielded by the Solar and Prox Cen spectra (blue and red lines respectively) against pressure ( Pa) on a log scale *left* column, and as a function of wavelength ($\mu$ m) for the Solar case *middle* column and Prox Cen case *right* column. Figure 18 shows the single plots of the dissociation rates of $NO_2 \rightarrow NO + O(^3P)$ and $NO_3 \rightarrow NO + O_2$ as yielded by the Prox Cen spectrum only (red line) on a linear scale against pressure ( Pa).

For the rates of $NO_2 \rightarrow NO + O(^3P)$ as a function of pressure, *top* row of Figure 17, the Prox Cen values (red) are compared to that of the Solar case (blue) and are generally much lower at all pressures for the Prox Cen case. The *top right* panel shows the photolysis rates at the TOA (blue), upper-mid atmosphere $\sim100$ Pa (green) and lower-mid atmosphere $\sim32000$ Pa (magenta), revealing a greater contribution to the photolysis for shorter wavelengths at the TOA compared to the Solar case (compare to Figure 7). For the Prox Cen case oxygen absorption at these shorter wavelengths impacts the photolysis higher up in the atmosphere, with the contribution from longer wavelengths being weaker, resulting in the steep decline of photolysis rates with increasing pressures, not seen in the Solar case. Figure 18, *top* panel, shows a zoom-in on the Prox Cen profile for this case. The small increase in rates at $\sim10^2$ Pa coincides with a peak in the temperature dependence of the cross section, which comes into effect at the longer UV wavelengths.





Interestingly for the case of NO$_3$ into NO and O$_2$, the Prox Cen case, Figure 18, *bottom* panel, shows the rates changing as a function of pressure in a noticeably different way to that of the Solar case (see Figure 7). The spectra of the rates as a function of pressure for wavelengths between 580-640 nm are shown at the TOA (blue), ∼140 Pa (green) and ∼7600 Pa (magenta) in the second row *right* panel of Figure 17. As there is very little absorption at these wavelengths, the spectral changes for different pressures is almost entirely due to the temperature dependence of the quantum yield. The TOA and 7600 Pa lines are essentially on top of each other because the temperature is about the same at these levels. The difference in the pressure dependence of the Solar and Prox Cen rates is due to the wavelength variation of this temperature dependence of the quantum yield. Essentially, for shorter wavelength flux, the quantum yield decreases as the temperature increases. Whereas, towards longer wavelengths, the quantum yield increases as the temperature increases. As the Prox Cen spectrum has a higher fraction of flux at longer wavelengths, compared to the Solar case, due to it being a cooler star, the rates are driven by the temperature of the atmosphere. Whereas, for the Solar case, the shorter wavelength flux dominates and the impact of the temperature dependence of the quantum yield is muted.

Figure 19 shows the comparison of Solar and Prox Cen rates for the remaining NOx species. These rates are dominated by wavelengths >200nm and are therefore significantly higher in the Solar case.

### 4.2.5 Organic

Figure 20 shows rates for the dissociation of formaldehyde, H$_2$CO into H and HCO *top* row, H$_2$CO→H$_2$+CO *middle* row and OCS→CO+S($^3$P) *bottom* row as yielded by the Solar and Prox Cen spectrum (blue and red line respectively) against pressure on a log scale *left* column, and as a function of wavelength ($\mu$ m) for the Solar case *middle* column and Prox Cen case *right* column at the TOA (blue) and ∼320 Pa (green).

When examining H$_2$CO, the differences are due to the increased FUV and EUV for the Prox Cen spectrum compared to that of the Solar case, leading to oxygen absorption in the upper atmosphere and a sharp decrease in rates with pressure, similar to behaviour displayed in the NO$_2$ case in Figure 17. This is demonstrated by the rates as a function of wavelength ($\mu$ m) at the TOA (blue) and ∼320 Pa (green), shown in the *right* panels of Figure 20, which are similar to the corresponding data for NO$_2$. Looking at the photolysis rates for OCS as a function of wavelength ($\mu$ m), *bottom* row of Figure 20, the rates between wavelengths of 200-300 nm vary somewhat more than those for the Solar case, likely caused by the commensurate noise in the input spectra across this range particularly around ∼200-210 nm.

Figure 21 shows the calculated rates as a function of pressure for the reaction CH$_3$→CH$_2$(1)+H *top* row and 4 dissociation rates of CH$_4$. Note that CH$_2$(1) is the methylene group where (1) refers to the excited singlet state. For the methyl radical-CH$_3$ extremely limited data are available (see Appendix A, Table A1), and only covered one band centred on 0.2155 nm with zero rates elsewhere.

Figures 22 and 23 display the dissociation rates of the same organic species displayed in Figure 11 for the Solar and Prox Cen spectra. The photolysis rates as produced by the Prox Cen spectrum for the species CH$_3$CHO, CHOCHO, CH$_3$COCH$_3$, HOCH$_2$CHO all display similar trends to that of OCS (see Figure 20).





Figure 24 shows the rates for photolysis reactions of $C_2H_2$, $C_2H_3$ and $C_2H_4$ as yielded by the Solar and Prox Cen spectra. For both $C_2H_2$ and $C_2H_4$, the photolysis rates for Prox Cen are higher than Solar at the top of atmosphere due to the greater contribution from Lyman-$\alpha$ wavelengths. At lower altitudes the contribution from the NUV dominates and Solar rates are

higher than for Prox Cen.

Figure 25 shows the rates for photolysis reactions of $C_2H_6$ as yielded by the Solar and Prox Cen spectra. These reactions are dominated by wavelengths around Lyman-$\alpha$ and are correspondingly higher for Prox Cen.

### 4.2.6 Other Exoplanet Species

In addition to the species already explored, some additional species are required for exoplanets where comparison rates under

an Earth-like $O_3$ profile are not available, including $H_2O$ which is detailed in Section 4.2.3. Therefore, in this section we simply provide our calculated rates as a reference for future studies. Figure 26 shows the rates for the dissociation of $CO + h\nu \rightarrow C +$ $O(^3P)$, $CO_2$ into CO and $O(^1D)$ and $CO_2$ into CO and $O(^3P)$, $HCN + h\nu \rightarrow H + CN$ and $NH_3 + h\nu \rightarrow NH_2 + H$ as yielded by the Solar and Prox Cen spectrum (blue and red line respectively) against pressure on a log scale *left* column, and as a function of wavelength ($\mu$ m) for the Solar case *middle* column and Prox Cen case *right* column at the TOA (blue) and $\sim$20 Pa (green).

The cross sections of $CO_2$ have a temperature dependence (see Table A1) and the effect of this is evident for $CO_2 \rightarrow CO +$ $O(^3P)$ where we see a protrusion indicating an increase in rates around $\sim$100 Pa. The Prox Cen rates for $CO_2 \rightarrow CO + O(^1D)$ are higher than the Solar rates due to the contribution around Lyman-$\alpha$. The short wavelengths are attenuated before arriving at $\sim$100 Pa which is why we do not see a similar peak. The threshold for the production of $O(^1D)$ is 167 nm and the quantum yield is zero below 50nm and therefore the flux supplied in the relevant wavelength range would be higher for Prox Cen than

that provided by the Solar spectrum. Similar reasoning can be applied to CO while HCN and $NH_3$ both have contributions from the NUV where the Solar flux is larger. For HCN the Lyman-$\alpha$ and NUV rates are balanced with Lyman-$\alpha$ dominating towards the top of the atmosphere and the NUV dominating below $\sim$100 Pa (similar to $C_2H_2$ and $C_2H_4$).

For ammonia ($NH_3$) similar to the case of $H_2CO$ and $NO_2$, as detailed in Sections 4.2.5 and 4.2.4, the longer UV wavelength contribution is smaller for Prox Cen than when the Solar spectrum is used. For the rates calculated with the Prox Cen spectrum

the shape of the rates as a function of pressure is mainly due to oxygen absorption of FUV flux at the top of the atmosphere, particularly around Lyman-$\alpha$. Whereas the photolysis rates calculated with the Solar spectrum are dominated by the NUV where ozone absorption lower in the atmosphere is the most significant factor.

## 5 Conclusions

Photochemistry is an important process in the atmospheres of planets, and therefore accurate photolysis schemes in models

are essential. In this paper, we first benchmark and test the Socrates photolysis scheme against the results of PhotoComp (Chipperfield et al., 2010) under an Earth atmosphere profile. The Socrates photolysis scheme generally compares well with the PhotoComp reference calculations. However, we also find the following:





- Significant differences can be present due to the adoption (or non-adoption) of temperature-dependent cross sections, e.g. $NO_2$, or quantum yields, e.g. $NO_3$. This can alter how the photolysis rates change through the atmosphere and possibly offset them.

- For some species, such as $O_2$ and $N_2O$, differences are caused by the inclusion or omission of EUV and shorter wavelengths altering rates towards the top-of-atmosphere. Within many of the spectra we see contributions from these shorter wavelengths which are particularly important around the strong Lyman-$\alpha$ emission line in the solar spectrum.

- In some cases we can find no direct cause of discrepancies between our rates and those of the PhotoComp references, therefore the cause is likely choices of input data, absorption coefficients, input spectra and model differences.

- In particular, much more work is required for the organic species as this is where the most significant discrepancies lie, which is partly due to uncertainties in choice of experimental data.

In Section 4.2, we then changed the input stellar spectrum to an M dwarf spectrum, but retained the same Earth-like atmospheric conditions and total incoming TOA flux as used for the calculations in Section 4.1. We find that the differences between the rates yielded from the solar spectrum versus the Proxima Centauri spectrum are accounted for generally by the higher levels of actinic flux below around 175nm, and lower levels at longer wavelengths in the Proxima Centauri spectrum and where the threshold of the photolysis pathways occurs within this wavelength region. In this sense, our results match the findings of Ridgway (2023). The variation as a function of wavelength of the input spectrum also affects the rates as a function of pressure through the atmosphere. For a number of species we find that the Prox Cen rates change more quickly as a function of pressure in the upper atmosphere due to a large contribution from EUV wavelengths sensitive to oxygen absorption. In contrast, rates from the Solar spectrum have a larger contribution from longer UV wavelengths which are sensitive to ozone absorption lower down in the stratosphere.

With the advent of new stellar input spectra including good coverage of the UV range via computational modelling alongside observations (e.g. Wilson et al., 2024; Linsky and Redfield, 2024), Socrates's ability to easily interchange the input stellar spectrum will be vital for exoplanet studies.

## 5.1 Future Work

A specific subset of species have been benchmarked and tested in this work. However, the ability to include other species such as halogenated species for Earth, as well as other species important for exoplanets and early Earth-like environments, such as sulphur and other hydrocarbon species, will be important for future studies.

We are currently performing a similar benchmarking exercise for species and conditions relevant to hot Jupiters. Hot Jupiters are Jovian planets in short-period orbits where tidal interactions lead to synchronised orbital and rotation periods, producing a dayside receiving constant and intense levels of irradiation (Showman and Guillot, 2002). Although there has been extensive work on the thermal chemistry of hot Jupiter atmospheres (Drummond et al., 2016; Zamyatina et al., 2024), there are very few benchmarks that exist for the photolysis rates of the relevant species under hot Jupiter atmospheric conditions. Different planet



environments covering different temperature regimes will require the aforementioned linear interpolation for temperature-dependent quantum yields and cross sections (see Section 2). This will be an evolving area as new data, especially high-temperature data for exoplanets, become available (e.g. Ni et al., 2025).

*Code and data availability.* The current version of Socrates is available from https://code.metoffice.gov.uk/trac/socrates under a BSD 3-clause licence. The exact version of the model used to produce the results used in this paper is available via Zenodo at
https://doi.org/10.5281/zenodo.16324116 (Adams et al., 2025), as are input data and scripts to run the model and produce the plots for all the simulations presented in this paper (Adams et al., 2025).

## Appendix A: Data Sources

The full details of all reactions and species used in our calculations, including our data sources for the absorption cross sections and quantum yields are shown in Table A1. The additional species: $CH_3$, $CH_4$, $C_2H_2$, $C_2H_3$, $C_2H_4$, $C_2H_6$, CO, HCN and $NH_3$
and their photolysis pathways use the same cross sections and quantum yields as Venot et al. (2012).



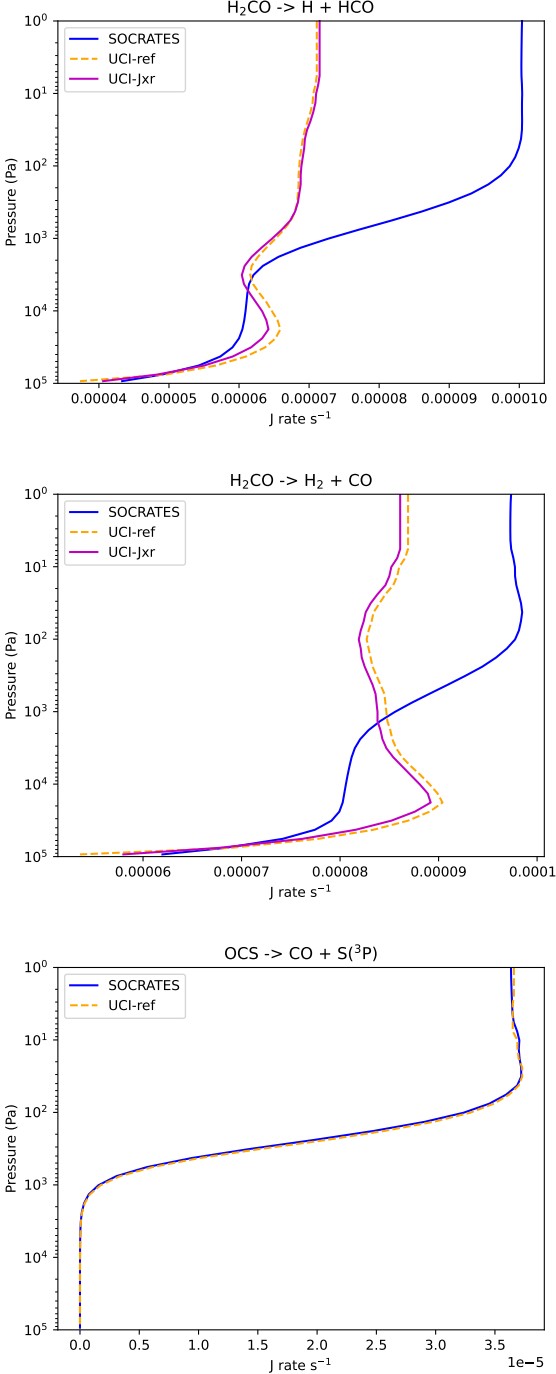

**Figure 10.** Photolysis rates of H$_2$CO→H+HCO (*top* panel), H$_2$CO→H$_2$+CO (*middle* panel) with the additional UCI-Jxr plot included (purple), and OCS→CO+S($^3$P) (*bottom* panel) as a function of pressure ( Pa).





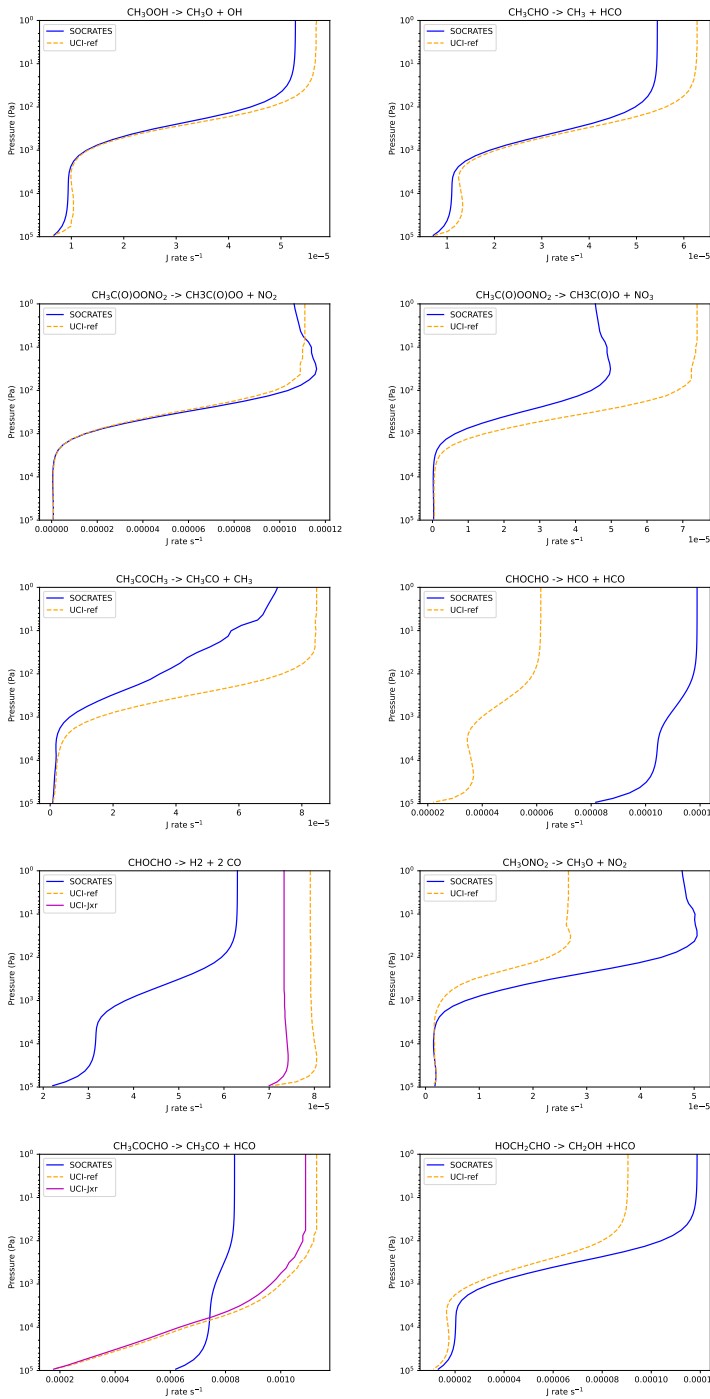

**Figure 11.** Photolysis rates for the remaining organic species as a function of pressure (Pa). Note that that species $CH_3C(O)OONO_2$ corresponds to PAN in the text.



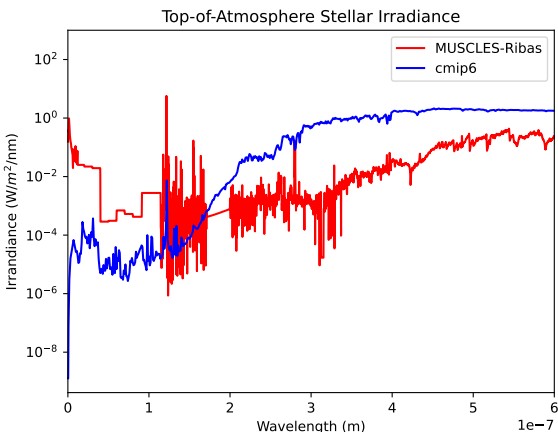

**Figure 12.** Figure based on Ridgway (2023)'s work showing the top-of-atmosphere stellar irradiance for the Solar CMIP6 spectrum at 1AU, compared with the combined MUSCLES-Ribas Prox Cen spectrum at $\sim 0.02$AU which arises out of keeping the total incoming flux the same.



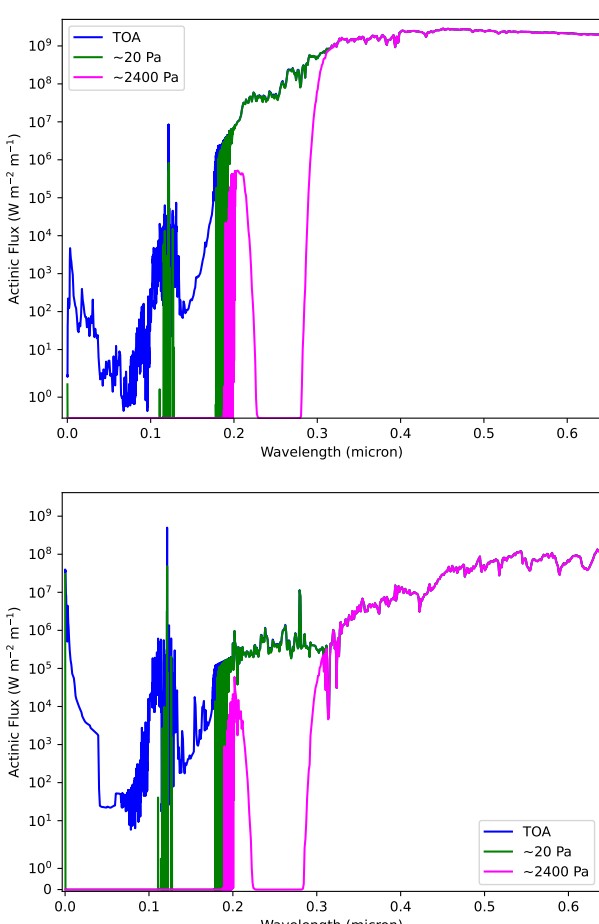

**Figure 13.** Actinic flux ($Wm^{-2}m^{-1}$) as a function of wavelength ($\mu m$) at three different levels, the top-of-atmosphere, upper mid-atmosphere (a pressure of $\sim$20 Pa) and lower mid-atmosphere (at a pressure of $\sim$2,396 Pa) corresponding to the ozone layer, shown by the solid blue, green, and magenta lines, respectively for both the Solar spectrum (*top* panel) and Prox Cen spectrum, (*bottom* panel).

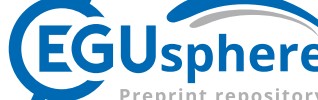



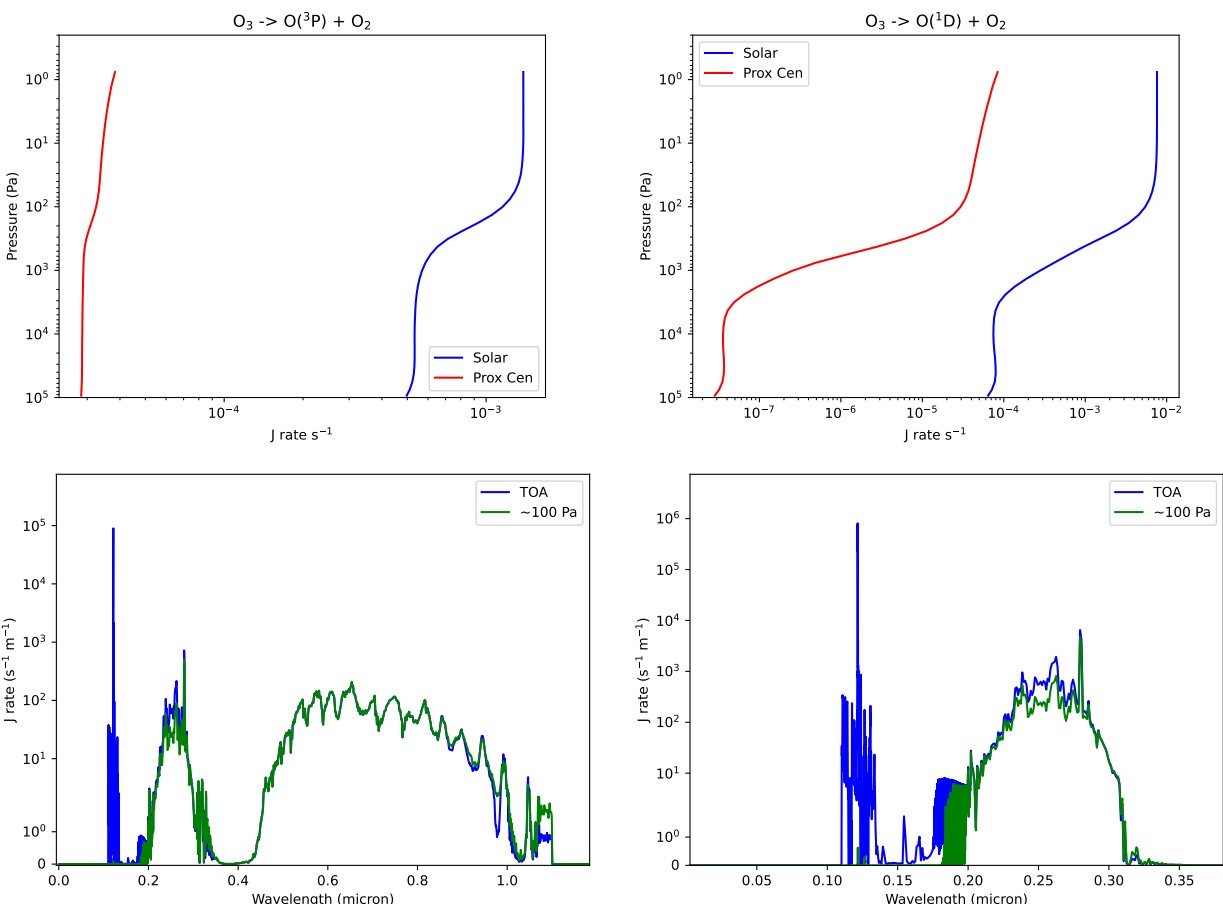

**Figure 14.** Photolysis rates on a log scale for the reactions $O_3 \rightarrow O_2 + O(^3P)$ and $O_3 \rightarrow O_2 + O(^1D)$ as yielded by the Solar and Prox Cen spectra (blue and red lines respectively) as a function of pressure (Pa, *top* panels). The photolysis rates for just the Prox Cen spectrum are shown as a function of wavelength ($\mu$m) in the *bottom* panels at the TOA and a pressure of $\sim$100 Pa (blue and green lines respectively).

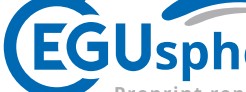

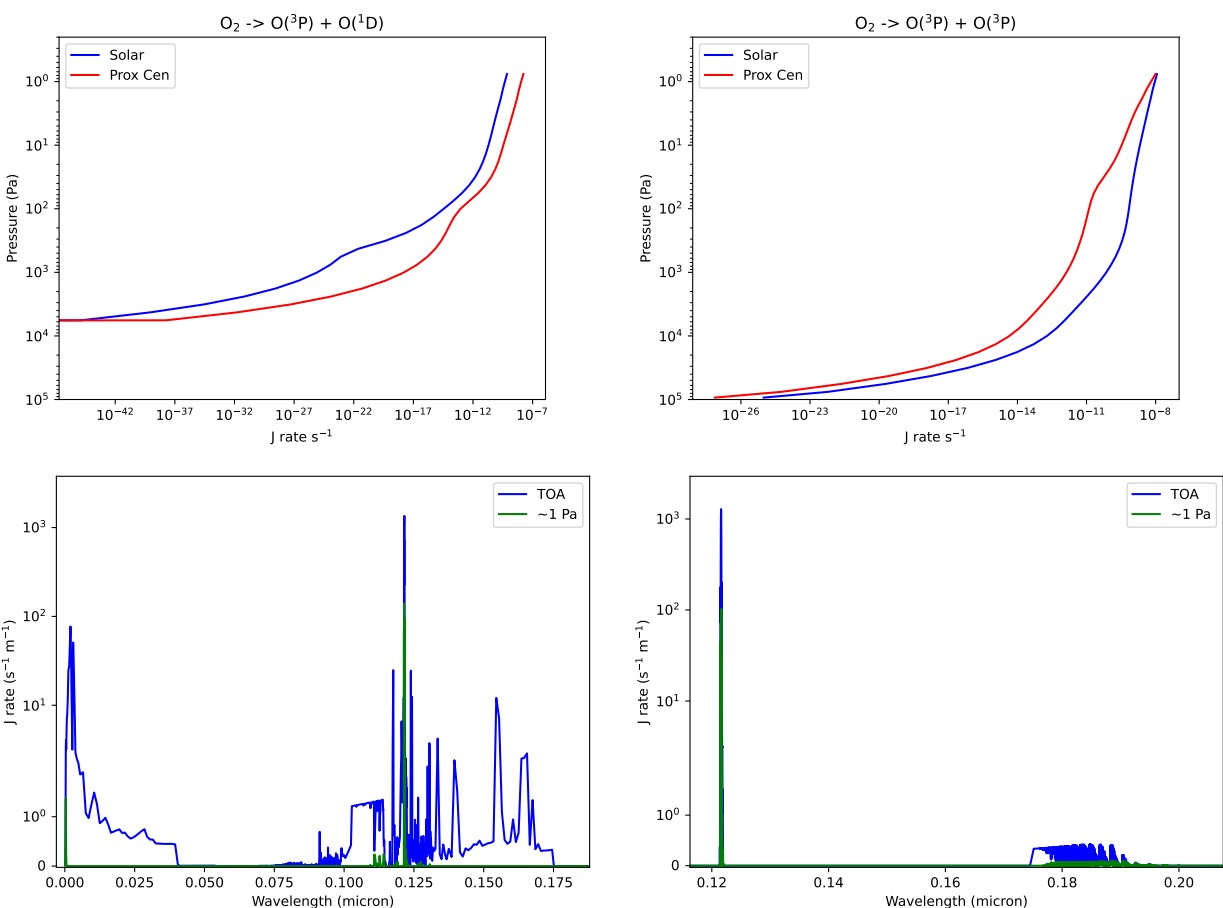

**Figure 15.** Photolysis rates on a log scale for the reactions $O_2 \rightarrow O(^3P) + O(^1D)$ and $O_2 \rightarrow O(^3P) + O(^3P)$ as yielded by the Solar and Prox Cen spectra (blue and red lines respectively) as a function of pressure (Pa, *top* panels). The photolysis rates for just the Prox Cen spectrum are shown as a function of wavelength ($\mu$ m) in the *bottom* panels at the TOA and a pressure of $\sim$1 Pa (blue and green lines respectively).





**Figure 16.** Three dissociation reactions of $H_2O$ and the dissociation of $H_2O_2$ as yielded by the Solar and Prox Cen spectrum (blue and red line respectively) against pressure ( Pa) on a log scale *left* column, and as a function of wavelength ($\mu$ m) for the Solar case *middle* column and Prox Cen case *right* column at the TOA (blue) and $\sim$20 Pa (green) for $H_2O$ and TOA (blue) and $\sim$320 Pa (green) for $H_2O_2$.



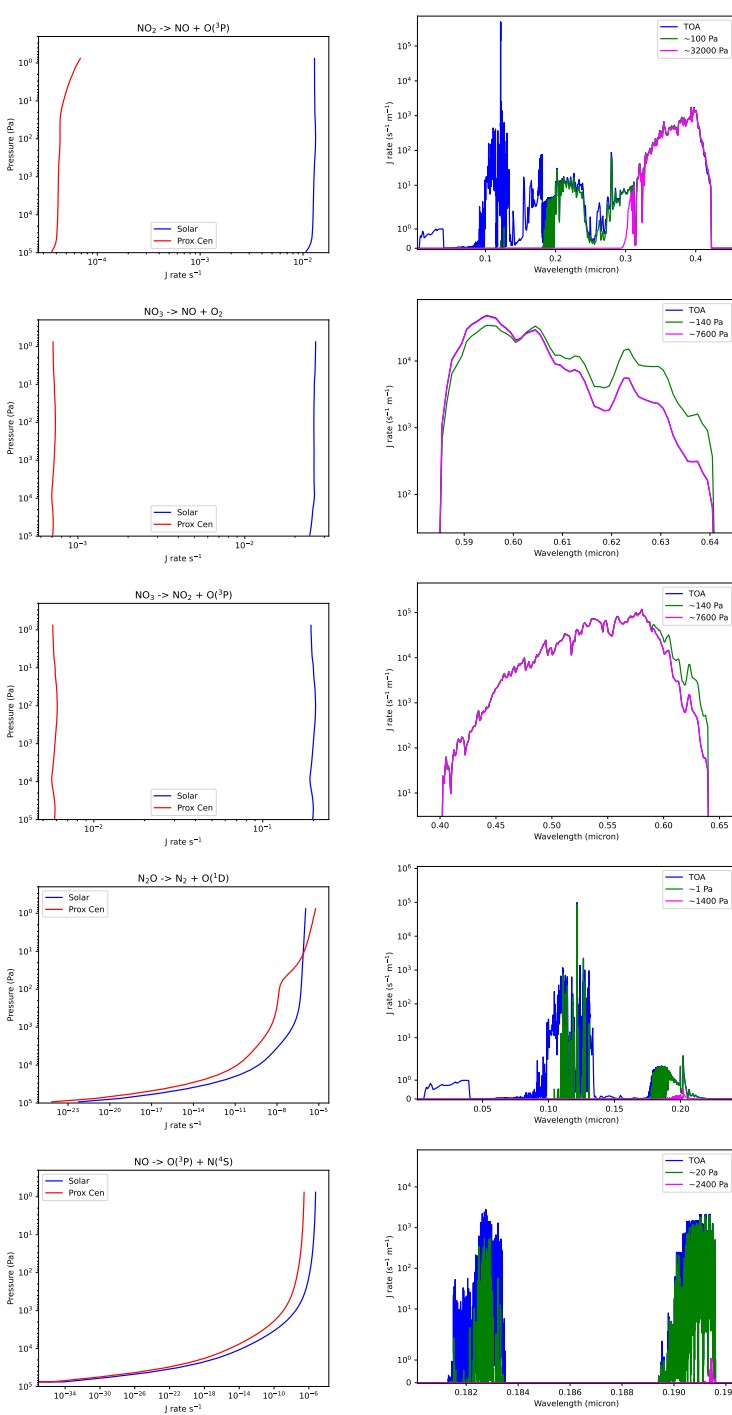

**Figure 17.** Photolysis rates of the reactions for the NOx species: $NO_2$, $NO_3$, $N_2O$ and $NO$ as a function of pressure (Pa, *left* panel), and as a function of wavelength ($\mu$m, *right* panel) in the same format as Fig. 7.





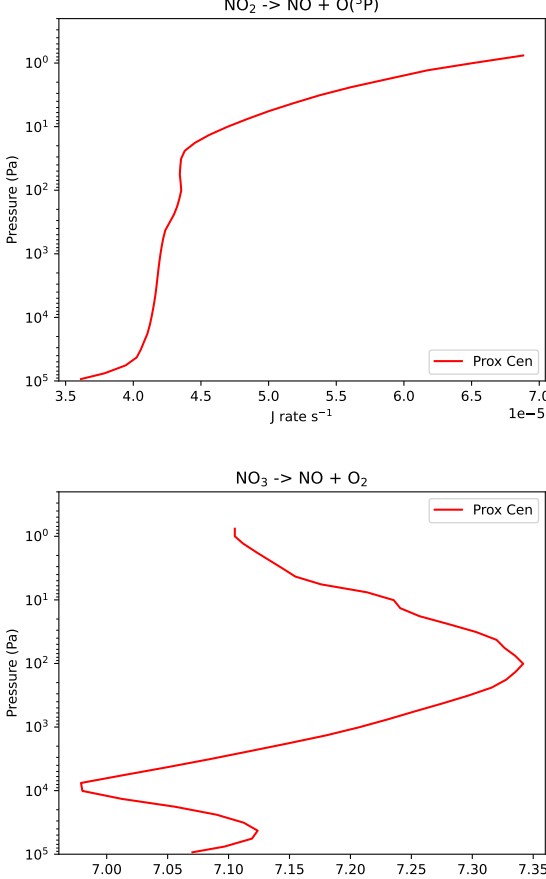

**Figure 18.** Dissociation rates of $NO_2 \rightarrow NO + O(^3P)$ and $NO_3 \rightarrow NO + O_2$ as yielded by the Prox Cen spectrum only (red line) on a linear scale against pressure ( Pa) .





**Figure 19.** Dissociations rates for the remaining NOx species as yielded by the Solar and Prox Cen spectrum (blue and red line respectively) against pressure ( Pa) on a log scale *left* column, and as a function of wavelength ($\mu$ m) for the Solar case *middle* column and Prox Cen case *right* column at the TOA (blue) and $\sim$20 Pa (green) in the same format as Fig. 16







**Figure 20.** Dissociations rates for $H_2CO$ and OCS as yielded by the Solar and Prox Cen spectrum (blue and red line respectively) against pressure on a log scale *left* column, and as a function of wavelength ($\mu$ m) for the Solar case *middle* column and Prox Cen case *right* column at TOA (blue) and $\sim$320 Pa (green) in the same format as Fig. 16.





**Figure 21.** Dissociation rates as a function of pressure for species $CH_4$ and $CH_3$ as yielded by the Solar and Prox Cen spectrum (blue and red line respectively) against pressure (Pa) on a log scale *left* column, and as a function of wavelength ($\mu$m) for the Solar case *middle* column and Prox Cen case *right* column at the TOA (blue) and $\sim$20 Pa (green).





**Figure 22.** Dissociation rates for further organic species as yielded by the Solar and Prox Cen spectrum (blue and red line respectively) against pressure ( Pa) on a log scale *left* column, and as a function of wavelength ($\mu$ m) for the Solar case *middle* column and Prox Cen case *right* column at the TOA (blue) and $\sim$20 Pa (green) in the same format as Fig. 16





**Figure 23.** Dissociation rates for further organic species as yielded by the Solar and Prox Cen spectrum (blue and red line respectively) against pressure (Pa) on a log scale *left* column, and as a function of wavelength ($\mu$m) for the Solar case *middle* column and Prox Cen case *right* column at the TOA (blue) and $\sim$20 Pa (green), and $\sim$2400 Pa (green) in the same format as Fig. 16





**Figure 24.** Dissociation rates for $C_2H_2$, $C_2H_3$ and $C_2H_4$ as yielded by the Solar and Prox Cen spectrum (blue and red line respectively) against pressure (Pa) on a log scale *left* column, and as a function of wavelength ($\mu$m) for the Solar case *middle* column and Prox Cen case *right* column at the TOA (blue) and $\sim$20 Pa (green) in the same format as Fig. 16.







**Figure 25.** Dissociation rates for $C_2H_6$ as yielded by the Solar and Prox Cen spectrum (blue and red line respectively) against pressure ( Pa) on a log scale *left* column, and as a function of wavelength ($\mu$ m) for the Solar case *middle* column and Prox Cen case *right* column at the TOA (blue) and $\sim$20 Pa (green) in the same format as Fig. 16





**Figure 26.** Dissociation rates for the remaining species relevant to exoplanets as yielded by the Solar and Prox Cen spectrum (blue and red line respectively) against pressure (Pa) on a log scale *left* column, and as a function of wavelength ($\mu$m) for the Solar case *middle* column and Prox Cen case *right* column at the TOA (blue) and ~20 Pa (green) in the same format as Fig. 16





| Species | Reaction | Cross Section Sources | Quantum yield Sources |
|---|---|---|---|
| Ozone | $O_3 + h\nu \rightarrow O(^3P) + O_2$ | Based on JPL 19-5 recommendations: 110 nm-185 nm (298 K) Mason et al. (1996) , 185-233 nm (298 K) Molina and Molina (1986), 233-244 nm (298 K) Burrows et al. (1999), 195 nm-244 nm (218 K) Malicet et al. (1995). Between 244 nm-346 nm: HITRAN 2020 data Gordon et al. (2022) at 6 temperatures. Between 346 nm-830 nm Brion et al. (1998) at 295 K (JPL 19-5 recommendation), 830-1100 nm Serdyuchenko et al. (2011) at 11 temperatures | Matsumi et al. (2002) |
| Ozone | $O_3 + h\nu \rightarrow O(^1D) + O_2$ | As above | Matsumi et al. (2002) |
| Oxygen | $O_2 + h\nu \rightarrow O + O$ | 0.04-4.48 nm Henke et al. (1993), 4.53-102.70 nm Fennelly and Torr (1992), 103.1-107.7 nm Matsunaga and Watanabe (1967), 107.93-108.64 nm Watanabe and Marmo (1956), 108.75-114.95 Ogawa and Ogawa (1975), 115-179 nm Lu et al. (2010), 179.21-202.58 nm Yoshino et al. (1992), 203-204 nm Hébrard (priv. comm. 2022), 205-240 nm Burkholder et al. (2020), 240.89-294.03 nm Fally et al. (2000) | EUV: Fennelly and Torr (1992), <65 nm enhancement factors: Solomon and Qian (2005), around Lyman-$\alpha$: Lacoursiere et al. (1999) |
| Hydrogen Peroxide | $H_2O_2 + h\nu \rightarrow OH + OH$ | JPL 19-5 Burkholder et al. (2020) | IUPAC Atkinson et al. (2004) |
| Nitrogen Dioxide | $NO_2 + h\nu \rightarrow NO + O(^3P)$ | 6-184 nm Au and Brion (1997), 185-200 nm Bass et al. (1976), 200-237 nm Merienne et al. (1995), 238-667 nm Vandaele et al. (1998) at 220 K & 298 K | JPL 19-5 Burkholder et al. (2020) |





| | | | |
|---|---|---|---|
| Nitrate | $NO_3 + h\nu \rightarrow NO + O_2$ | Based on the recommendations of JPL 19-5 Burkholder et al. (2020), 400-691 nm Sander (1986) | JPL 19-5 Burkholder et al. (2020) |
| Nitrate | $NO_3 + h\nu \rightarrow NO_2 + O(^3P)$ | As above | JPL 19-5 Burkholder et al. (2020) |
| Nitrous Oxide | $N_2O + h\nu \rightarrow N_2 + O(^1D)$ | Hébrard (priv. comm. 2022): Au and Brion (1997), Hubrich and Stuhl (1980), Selwyn et al. (1977), Hubrich and Stuhl (1980) | JPL 19-5 Burkholder et al. (2020) |
| Nitric Oxide | $NO + h\nu \rightarrow O(^3P) + N(^4S)$ | Chang et al. (1993), Iida et al. (1986), XABC line list data Wong et al. (2017) as sourced from Exomol Tennyson et al. (2016) | |
| Dinitrogen pentoxide | $N_2O_5 + h\nu \rightarrow NO_2 + NO_3$ | Hébrard (priv. comm. 2022): Osborne et al. (2000), Yao et al. (1982), Harwood et al. (1998) | IUPAC Atkinson et al. (2004) |
| Nitrous Acid | $HONO + h\nu \rightarrow OH + NO$ | 184-271 nm Kenner et al. (1986), 291-404 nm Stutz et al. (2000) 296 | IUPAC Atkinson et al. (2004) |
| Nitric Acid | $HNO_3 + h\nu \rightarrow NO_2 + OH$ | Based on the recommendations of JPL 19-5: Burkholder et al. (1993) 186-350 nm at 240 K, 260 K, 280 K, 298 K, and Suto and Lee (1984) 105 nm-225 nm at 298 K | IUPAC Atkinson et al. (2004) |
| Peroxynitric acid | $HO_2NO_2 + h\nu \rightarrow OH + NO_3$ | Hébrard (priv. comm. 2022) | JPL 19-5 Burkholder et al. (2020) |
| Formaldehyde | $H_2CO + h\nu \rightarrow H + HCO$ | 6-115 nm Cooper et al. (1996), 116 nm-180 nm Suto et al. (1986), 181 nm-225 nm Hébrard (priv. comm. 2022), 226 nm-376 nm Meller and Moortgat (2000) at 223 K & 298 K | JPL 19-5 Burkholder et al. (2020) at standard pressure (1 atmosphere) and 300 K |
| Formaldehyde | $H_2CO + h\nu \rightarrow H_2 + CO$ | As above | JPL 19-5 Burkholder et al. (2020) at standard pressure (1 atmosphere) and 300 K |
| Carbonyl sulfide | $OCS + h\nu \rightarrow CO + S(^3P)$ | JPL 19-5 Burkholder et al. (2020) | JPL 19-5 Burkholder et al. (2020) |
| Methyl- | | | |





| | | | |
|---|---|---|---|
| hydroperoxide | $CH_3OOH + h\nu \rightarrow CH_3O + OH$ | JPL 19-5 Burkholder et al. (2020) | JPL 19-5 Burkholder et al. (2020) |
| Acetaldehyde gas | $CH_3CHO + h\nu \rightarrow CH_3 + HCO$ | JPL 19-5 Burkholder et al. (2020) | JPL 19-5 Burkholder et al. (2020) |
| Poly-acrylonitrile | $PAN + h\nu \rightarrow CH_3C(O)OO + NO_2$ | Based on the recommendations of JPL 19-5: Talukdar et al. (1995) at 250 K, 273 K, 298 K | JPL 19-5 Burkholder et al. (2020) |
| Poly-acrylonitrile | $PAN + h\nu \rightarrow CH_3C(O)O + NO_3$ | As above | JPL 19-5 Burkholder et al. (2020) |
| Acetone | $CH_3COCH_3 + h\nu \rightarrow CH_3CO + CH_3$ | Based on the recommendations of JPL 19-5: Gierczak et al. (1998) with parameterisations revised by Burkholder et al. (2020) at temperatures 235 K, 254 K, 263 K, 280 K, 298 K | T-dependence uses formulation from Blitz et al. (2004) using tropospheric pressures 154, 273.8, 487, 866 hPa for the temperatures 218, 248, 273 and 295 K respectively. |
| Glyoxal | $CHOCHO + h\nu \rightarrow HCO + HCO$ | JPL 19-5 Burkholder et al. (2020) | JPL 19-5 Burkholder et al. (2020) |
| Glyoxal | $CHOCHO + h\nu \rightarrow H_2 + 2CO$ | JPL 19-5 Burkholder et al. (2020) | JPL 19-5 Burkholder et al. (2020) |
| Methylnitrate | $CH_3ONO_2 + h\nu \rightarrow CH_3O + NO_2$ | Based on the recommendations of JPL 19-5: 190 nm-235 nm Taylor et al. (1980), 236 nm-334 nm Talukdar et al. (1997) at temperatures 240 K, 260 K, 280 K, 298 K, 320 K, 340 K and 360 K | JPL 19-5 Burkholder et al. (2020) |
| Methylglyoxal | $CH_3COCHO + h\nu \rightarrow CH_3CO + HCO$ | JPL 19-5 Burkholder et al. (2020) | JPL 19-5 Burkholder et al. (2020) |
| Glycolaldehyde | $HOCH_2CHO + h\nu \rightarrow CH_2OH + HCO$ | JPL 19-5 Burkholder et al. (2020) | JPL 19-5 Burkholder et al. (2020) |
| Water | $H_2O + h\nu \rightarrow OH(X_2\pi) + H$ | Hébrard (priv. comm. 2022) Collated: Chan et al. (1993), Mota et al. (2005), Fillion et al. (2004), Ranjan et al. (2020) | JPL 19-5 Burkholder et al. (2020) |





| Water | $H_2O + h\nu \rightarrow O(^1D) + H_2$ | Hébrard (priv. comm. 2022): As above | JPL 19-5 Burkholder et al. (2020) |
|---|---|---|---|
| Water | $H_2O + h\nu \rightarrow O(^3P) + H_2$ | Hébrard (priv. comm. 2022): As above | JPL 19-5 Burkholder et al. (2020) |
| Carbon Dioxide | $CO_2 + h\nu \rightarrow O(^3P) + CO$ | 1-114 nm at 300 K: Venot et al. (2012), 115-800 nm for 150 K-800 K: Venot et al. (2018) | Venot et al. (2012) |
| Carbon Dioxide | $CO_2 + h\nu \rightarrow O(^1D) + CO$ | As above | Venot et al. (2012) |

Table A1: Species, reactions and sources.

*Author contributions.* SA led the work, collated the input data and performed the calculations as well as leading the writing of the manuscript. JM supported the development of the input files and calculations using Socrates, as well as aiding in the development of the manuscript. NM provided overall guidance, supervision and resources for the work, and aided in the scientific analysis and development of the manuscript. MTM provided direct support with development of optical properties and the use of Socrates, alongside helping with the scientific analysis. EH provided expertise and guidance in the collation of the input data and photolysis reactions.

*Competing interests.* The authors have no competing interests.

*Acknowledgements.* We would like to acknowledge Martyn Chipperfield for providing data for this work. Sophia Adams was supported by a Black British Researchers Scholarship at the University of Exeter (REF: 4727), made possible through generous alumni donations. This research was supported by a (UKRI) Future Leaders Fellowship MR / T040866 / 1 and a Small Award from the Science and Technology

Facilities Council for Astronomy Observation and Theory [ST / Y00261X / 1]. Material produced using Met Office Software. We acknowledge use of the Monsoon2 system, a collaborative facility supplied under the Joint Weather and Climate Research Programme, a strategic partnership between the Met Office and the Natural Environment Research Council. M.T.M. acknowledges funding from the Bell Burnell Graduate Scholarship Fund, administered and managed by the Institute of Physics (BB005). For the purpose of open access, the author has applied a Creative Commons Attribution (CC BY) licence to any Author Accepted Manuscript version arising from this submission.



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
