# Peer review of "Benchmarking Photolysis Rates: Species for Earth and Exoplanets"

_EGUsphere, 2025_

## Author Comment (AC1)

Title: Benchmarking Photolysis Rates with Socrates (24.11): Species for Earth and Exoplanets

Author(s): Sophia Adams et al.

MS No.: egusphere-2025-2908

MS type: Model evaluation paper

*Dear reviewer,*

*Thank you very much for your comments. They have been very helpful and constructive and we believe have greatly improved and clarified the manuscript. Below we have listed all the comments, with our responses in* red *and italic. The corresponding updates in the manuscript are shown in* **bold** *text.*

*Kind regards,*
*Sophia Adams and co-authors*

This paper calculates photolysis rates using Socrates and evaluates the results through a comparison with two other reference models, UCI-ref and UCI-Jxr. The evaluation was conducted using both Earth's atmosphere, and an example exoplanet. The absorption cross-sections and quantum yields used were mostly based on expert panel reviews from IUPAC and JPL where available, with other sources used when necessary. The paper is a useful contribution to the field, especially as absorption cross-sections and quantum yields have been updated through this work, however more work needs to be done to extend the analysis of the discrepancies between the models, and improve consistency and readability before it can be published. Additionally, the quality of many of the figures (and their captions) does not seem to be up to GMD standards and should be improved.

Primarily, there are too many technical mistakes. Although many of them are minor, their quantity indicates a more thorough read through and edit is required by the authors (in addition to correcting those I have noticed in the 'Technical corrections' section).

A few sections mention the rates of reactions (lines 227,234), but there is no reference of how this was obtained, where are the quantum yields and absorption cross-sections from? Even if this is mentioned in Table A1, it should be referred to in the text.

Additionally, the analysis of the discrepancies between the models is missing depth. Is this work considered better than UCI (which may have quantum yields/absorption cross-sections that are not up-to-date)? Or is UCI already the standard, and this comparison is just to check the reliability of this work? Does UCI use pressure and temperature dependent quantum yield? It is

unclear from reading your paper. Differences between the two photolysis models should be explored further.

*Thank you very much for these comments. The technical mistakes have now been fixed. We address each point below. Mentions of where our data was obtained is specified within the text as well as Table A1.*

*We have provided a more in depth analysis behind the discrepancies. We have also revised and recalculated new photolysis rates based on previously missed temperature dependent cross sections which explained some of the difference for species: $HNO_3$, $N_2O$, $N_2O_5$ and $H_2O_2$.*

Finally, figures are not of a high quality and should be improved generally. Here are some specific examples, where detail, readability and consistency are lacking:

- Any mention of molecules should be correctly written i.e. '$O_2$' not 'O2' in Figure 2.

*Thank you, this has now been fixed.*

- Ensure that the labelling is consistent: in most figures it should be 'μm' instead of 'micron', and 'J rate $(s^{-1})$' instead of 'J rate $s^{-1}$'

*All labels have been changed to "$J (s^{-1})$" and all wavelengths are reported in nm.*

- Many caption descriptions are not detailed enough to stand alone. Some examples: Figure 3 should have clearer references to the legend used in the figure, Figures 4 and 5 bottom row have no mention of where that data comes from (along with all other figures that include a J rate against wavelength), Figure 7 caption is missing detail (legend not explained)

*Caption descriptions have been adjusted to provide more useful information.*

- Some of the figures are misaligned (such as Figures 7 and 16, but there are many examples)

*Figures have been adjusted so that they are all aligned.*

- The text and numbers are very small, should be increased for visibility

*The font size of the titles, axis ticks and legends have all been increased.*

**Specific Comments:**

Line 176: Why was Part 1a used, and not another set-up? Can you provide a sentence or two to explain? Can you discuss how only looking at one very specific scenario 'clear sky, no aerosols, high sun (Solar Zenith Angle, SZA = 15°) over the ocean' will affect the results?

*We only use Part 1a because other tests conducted by Photocomp were to test radiative transfer accuracy but this has been rigorously tested in Socrate's case so was not needed. The explanation in the paper is given as follows:* **"The PhotoComp study also included tests of the accuracy of the actinic flux calculations for different atmospheric compositions. However, the accuracy of the Socrates radiative transfer calculations has been extensively validated previously for both Earth (Pincus 2020) and exoplanets (Amundsen 2014), therefore we restrict our work here to benchmarking the photolysis rates only."**

Lines 226,247: You mention that there are differences due to input data, do you know why absorption cross-sections and quantum yields are different? Is one more up-to-date, do they use a different source?

*Yes, we are using more up-to-date cross sections. This has been clarified in the paper:* **"The UCI-ref model is based on JPL recommended cross-sections from 2010 while Socrates is using an updated temperature dependent cross-section from HITRAN 2020."**

Lines 294-295: Can you find the cause of the discrepancy, instead of listing potential causes

*We hadn't included the effect of temperature dependent cross sections. This has now been rectified, and the results match more closely with the reference model.*

Line 306: 'which the reference models may not' is there no way to know for sure?

*Thank you for your comment. Our previous results did not take into account temperature dependence in the cross sections. We have recalculated based on this new input data and provided new results. The $H_2O_2$ section has been rewritten with this considered and we simply state exactly what input data we are using.*

Lines 320-324: Can you add a figure (potentially in the appendix) demonstrating the effect of the pressure dependence on a J rate vs wavelength plot, at different pressures? This would be

similar to Figure 6 bottom panel, but showing the rate with and without pressure dependence of quantum yield? It would be good to do this for formaldehyde too

*Thank you for your suggestion. However, Socrates does not currently have the functionality to consider pressure dependence in the quantum yield. Future work to include this functionality in Socrates will be undertaken and a subsequent analysis on the impact on the results will be undertaken.*

Line 330: You mention there is a great deal of uncertainty with the input data, can you expand on this and provide examples? Especially as this is mentioned again in the last conclusion point

*For example, we expand upon the uncertainty for $CH_3ONO_2$ and provide the following explanation: **"The JPL 19-5 report does not provide recommended values of the quantum yield but reports conflicting values measured at particular wavelengths. We use a quantum yield of 1 for wavelengths $> 248 nm, 0.91 for wavelengths 241 - 248 nm and 0.7 for wavelengths < 241 nm. However the chosen limits are fairly arbitrary and likely to be the cause of the discrepancy between the Socrates and UCI-ref photolysis rates."***

**Technical Corrections:**

Generally:

- Fast-JX: capitalise the 'X' throughout the text

*This has been amended throughout the paper.*

- Molecule names should not be capitalised (formaldehyde lines 303 and 304, glyoxal line 326, carbonyl sulfide)

*The molecule names are no longer capitalized.*

Line 17: units should be separated with a space '25 km'

*A space has been added.*

Line 20: 'At the surface, UV irradiation has implications…' remove comma

*The comma has been removed.*

Line 24: This sentence is confusing, what do you mean by 'elements'?

*Thank you for your comment. This sentence has been reworded as follows:* **"Work exploring the cycling of ozone has been performed which demonstrated features such as formation of secondary ozone layers, and shielding from flaring caused by ozone build--up from previous flares"**

Line 29-31: This sentence is very hard to read and understand, please rework

This section had been reworded for clarity. The sentence now reads**: 'In order to calculate photolysis rates, we need information on the absorption cross section of the species involved, the quantum yield of the reactions (i.e. the branching ratio indicating which particular photolysis pathway is most probable), the spectrum of the incoming irradiance from the star at the top of the atmosphere and a treatment of the radiative transfer to determine the resultant actinic flux at a given atmospheric layer.'**

Line 90: there is an extra space after 'remembered'

*This section has been reworded therefore this is no longer an issue.*

Line 154: EUV was mentioned for the first time without a definition. Wavelength range covered in extreme UV category would also be useful

*EUV has now been defined as* **"extreme UV (EUV, <121 nm)"**

Line 202: What is 'N $^3$', is that a mistake?

*The "3" subscript was to indicate a footnote. This has been reformatted to avoid confusion.*

Line 209: 'O($^3$P)' not just '($^3$P)'

*This has been fixed.*

Figure 8 caption: Needs space between 'pressure( Pa)' should be 'pressure (Pa)'

*A space has been added.*

Line 259: I think it should say '400-420 nm' not '300-420 nm'

*This was a mistake and now reads as '400-420 nm'.*

Line 319: the word 'significant' should only be used if it is statistically significant. Is that the case here?

*Yes, it is the case where it is statistically significant here.*

Line 344: FUV not defined

*FUV had now been defined- ; The wavelength ranges have been defined **as "far UV (FUV, 121 - 200 nm) "***

Line 345: space needed '121.6nm'

*A space has been added.*

Line 359: NUV not defined, Figure 4 missing capital

*This has now been defined as "near-UV (NUV,  ~200-400 nm)" and the missing capital has been added.*

Line 382: 'and so similar to'?

*This section has been reworded and so this grammatical error is no longer an issue.*

Line 412: 200nm, space needed

*A space has been added.*

Line 459: 50nm, space needed

*A space has been added.*

Table A1: Table caption should be above a table, not below. More description of the table is needed in the caption

*The caption is now above the table, and more description of the table has been included.*

---

## Author Comment (AC2)

Title: Benchmarking Photolysis Rates with Socrates (24.11): Species for Earth and Exoplanets

Author(s): Sophia Adams et al.

MS No.: egusphere-2025-2908

MS type: Model evaluation paper

*Dear reviewer,*

*Thank you very much for your comments. They have been very helpful and constructive and we believe have greatly improved and clarified the manuscript. Below we have listed all the comments, with our responses in red and italic. The corresponding updates in the manuscript are shown in **bold** text.*

*Kind regards,*
*Sophia Adams and co-authors*

This paper reports predicted photolysis rate constants for the modern terrestrial atmosphere using the Socrates radiation scheme and compares them to CCMVal PhotoComp 2011. The Socrates model is then used to predict photolyis rates for the Earth if it were around Proxima Centauri at 0.02 AU, which is taken as an example of an Earth-like exoplanet in the habitable zone around an M dwarf star. The availability of values for exoplanet studies are novel and, if valid, of interest and value to the field. The significance and quality of the work is currently limited, however, as the authors provide little in the way of in-depth explanation for the discrepancies between their solar values and those of PhotoComp. This makes it difficult to be certain that this work is valid, let alone an improvement on previous work. They often ascribe differences between their results and reference models to "model differences" without clarification of whether these differences represent advances in the measurements of photolysis cross-sections, rate constants, quantum yields etc. and therefore provide a significant improvement to the PhotoComp models, or if their model is missing some key features or data.

The paper is generally difficult to read: figures are unclear and need significant improvements (detailed below) before the paper would be suitable for publication. In addition, the language used in the paper is often highly informal and overly verbose, with run-on and grammatically incorrect sentences appearing throughout. I have provided some examples of ways particularly confusing sentences could be reworded to be clearer and more direct, but the paper generally needs significant revisions to improve the clarity and formality of the language used.

Recurring and general issues:

- The figures generally need much improvement. Axis labels are too small to read, chemical species are often not formatted correctly (missing subscripts), arrows are not drawn correctly (using -> instead of → or similar), wavelength units are often given as "micron" rather than "μm". Many axis labels also fail to differentiate units from titles, e.g. "J rate s$^{-1}$", and need brackets or other means. The readability of plots would be improved by the inclusion of titles or other labels for each subplot, especially in the second half of the paper when rates using solar spectra and Proxima Centauri spectra are shown as separate subplots on the same figure, with no immediate indication of which is which. Line labels in figure legends are unclear and inconsistently formatted, with labels alternating between, for example, "SOCRATES" and "SOC". Other labels, for example "SOC no T QY" are entirely unintelligible without consulting the main text. Figure captions need additional details for the figures to be understandable without referring back to the main text of the paper. The authors could also consider providing row and column headings for figures with multiple subplots to make it clear that each row applies to the same reaction, and each column shows rate as a function of a different variable/for a different stellar spectrum. The use of subplot numbering or letters should also be considered (Figure Xa, b, etc.) to avoid the current reliance on "top row", "second row" etc. currently needed to distinguish between subplots. Finally, though this is a subjective point, the authors could consider switching the colours for solar and Proxima Centauri spectra. Using red for the warmer star (the sun) and blue for the cooler star (Proxima Centauri) may make the plots more instinctively readable.

*Thank you for the suggestion. We have increased the font size of the axis label, fixed the missing subscripts, drawn the arrows correctly, fixed all the units consistently in 'nm', and added brackets around units from titles. We have improved the readability by including titles, headings and labels in each subplot. All the legends and labels are formatted consistently. Figure captions have been updated to include additional background.*

*Regarding the last comment, we agree that using red for the warmer star (the Sun) and blue for the cooler star (Proxima Centauri) could be more intuitive. However, to maintain consistency with our original figure styling and with other plots in the manuscript, we have chosen to keep the current colour scheme. We have, however, clarified the legend and caption to more clearly distinguish the two spectra, so that the identification of each star is immediately apparent to the reader.*

- The model is at times referred to as "Socrates" and as "SOCRATES" at others. This should be consistent throughout the paper. Similarly, "PhotoComp" and "Photocomp" are both used throughout the paper, this should be consistent.

*Thank you for pointing this out. All the names have been formatted consistently.*

- Figures generally report wavelength in μm (though not always), whereas discussion in the text refers to wavelengths in nm. This conversion leads to mistakes when values in μm are reported in nm (e.g. "wavelength region 0.08-130 nm" (line 382), or "one band centred on 0.2155 nm" (line 427)). Units should be consistent throughout the paper, in both the text and the figures. The use of nm for both seems the most straightforward.

*Thank you for pointing this out. All units have been formatted consistently as nm.*

- Acronyms should be defined the first time they are used (e.g. EUV, GCMs)

*Thank you. All the acronyms have been defined the first time.*

When referring to subplots, the italicisation of top/bottom/left/right etc and row/column should be consistent (e.g. Figure 4 caption italicises both words, while the main text generally only italicises the first). Locations should always be in brackets in the main text, e.g. "$H_2CO$ into H and HCO top row, $H_2CO \rightarrow H_2 + CO$ middle row" (line 414) should be "$H_2CO$ into H and HCO (top row), $H_2CO \rightarrow H_2 + CO$ (middle row)".

*The italicisation of top/bottom/left and row/column has been made consistent*

- Throughout the paper, NOx, HOx, and Ox should instead be $NO_x$, $HO_x$, and $O_x$.

*Thank you for alerting us to this. NOx, HOx, and Ox have now consistently been changed to $NO_x$, $HO_x$, and $O_x$ throughout.*

- "J" is a photolysis rate, so "J rate" is cont technically correct. Labels should be either "J ($s^{-1}$)" or "photolysis rate ($s^{-1}$)", not "J rate $s^{-1}$".

*All labels have been changed to "J ($s^{-1}$)"*

- Values and units should be separated by a space (e.g. Figure 12 caption 0.02AU ahould be 0.02 AU)

*This has now been changed to 0.02 AU*

- The choice of pressure levels for comparison in Figure 6 onwards should be explained.

*Further explanation for the reasons behind the chosen pressure levels have been provided. For example, for Figure 6, the pressure level at the ozone layer is chosen because of how ozone absorption affects the results.*

- When comparing models or datasets in figures, the model that each spectrum or subplot is from must be labelled on the plot.

*This has now been rectified consistently throughout the paper.*

- "μ m", when used in the paper, is usually written with a space between the μ and m. It should be all one word and should not be partially italicised.

*All wavelength units have been consistently changed to nm throughout the paper.*

- From Figure 8 to the end of the purely solar section (4.1), figures do not contain plots of J as a function of wavelength. These appear to instead be grouped with the Proxima Centauri figures. Some explanation should be given for why the pressure the wavelength dependence of J is no longer relevant in this section, but was for earlier species and for the comparison of solar and Proxima Centauri spectra.

*Thank you for your comment. Any analysis of the both the Solar and Proxima Centuri spectra is in Section 4.2 which is why we do not include it there and this is noted in the paper.*

- The authors should consider whether the shortening of "Proxima Centauri" to "Prox Cen" is worthwhile, as it makes the manuscript slightly more difficult to read but does not decrease the number of words used and makes only a marginal change to the length of labels etc.

*All instances of "Prox Cen" have been changed to "Proxima Centauri".*

- Figures 4, 5, and 6 are shown with J vs. pressure on the top row and J vs wavelength on the bottom row. From Figure 7 onwards, this changes to J vs pressure on the left and J vs wavelength on the right. A consistent layout of similar figures would make the paper

easier to read. The authors could consider changing the layout of Figures 4, 5, and 6 to be consistent with later figures.

*The figures have been changed so that the layout is consistent*

Specific issues and suggestions:

- Line 20: "there are many other trace gases in the Earth's atmosphere that can undergo photolysis." Consider listing some examples here.

*Thank you for this suggestion. The sentence has been changed to:* **"There are also many other trace gases, such as organic molecules, in the Earth's atmosphere that can undergo photolysis"**

- Line 32: "Cross sections and quantum yield data are measured in laboratory experiments, with their subsequent recommended values collated in various literature sources." These can also be predicted from quantum calculations to fill in the gaps in laboratory data, is there a reason the authors have ignored such sources?

*Thank you for your comment. We have also used quantum yields predicted from quantum calculations but did not specify this previously. We have done so now and the sentence reads:* **"Cross sections and quantum yield data are measured in laboratory experiments or predicted from quantum calculations, with their subsequent recommended values collated in various literature sources. Photolysis models have been used to perform detailed 1D intercomparison studies and provide benchmark photolysis rates, given the input data, in the context of Earth, such as that of CCMVal PhotoComp 2011"**

- Line 37: The clause "...includes both a calculation of the radiative heating rates, and photolysis rates within a simulated atmosphere." Is grammatically incorrect and should be reworded to something like "includes calculation of both radiative heating rates and photolysis rates within a simulated atmosphere."

*Thank you for pointing this out. The sentence has been amended using your suggestion.*

- Line 41: Define "LFRic"

*This acronym has been defined-* **"LFRic (named after Lewis Fry Richardson)"**

- Line 42: Define "GCMs"

*This acronym has been defined as "Global Circulation Models"*

- Line 52: "wavelengths down to far and extreme UV" – give wavelength ranges for these regions

*The wavelength ranges have been defined **as "far UV (FUV, 121 - 200 nm) and extreme UV (EUV, <121 nm)"***

- Line 53: "$O_2$, $N_2$, and O absorption" implies these species are being absorbed, not that they are absorbing.

*This sentence has been amended and now reads as followed, **"Inclusion of the mesosphere and lower thermosphere requires a treatment of far UV (FUV, 121 - 200 nm) and extreme UV (EUV, <121 nm) wavelengths where absorption by $O_2$, $N_2$ and O become important"***

- Line 60: "high resolution" – specify the resolution or refer to a table or figure where it is shown

  *We now refer to Section 2.1.1.*

- Line 66: "Next" is not necessary here, and feels overly informal.

*This word has been removed.*

- Line 72: "Extra species relevant to exoplanets are included in an extra category" - specify the section here.

*We give the example of $H_2O$ and other hydrocarbons such as $C_2H_2$.*

- Figure 2: In addition to the general comments on figures above, "$cm^2 molecule^{-1}$" in caption should have a space in: $cm^2 molecule^{-1}$

*A space has been added to "$cm^2 molecule^{-1}$"*

- Line 84: "Essentially, ..." this sounds highly informal and should be removed or reworded

*This word has been removed.*

- Line 97: no comma after "where"

*This comma has been removed.*

- Line 98: The sentence starting "The flux in terms of ..." is unclear and should be reordered to something like "The flux is converted to units of photons $m^2$ $s^{-1}$ (A) by dividing by the energy of a photon with wavenumber of the midpoint of the sub-band."

*This sentence has been changed to the following: "**The flux is converted to units of photons $m^2$ $s^{-1}$ (A) by dividing by the energy of a photon with wavenumber of the midpoint of the sub-band.**"*

- Line 110: all non-unitless variables have their units defined here except wavelength, so units should be added for consistency

*Units have now been added.*

- Line 126: Some explanation should be provided for the switch from wavelength to wavenumber

*Thank you for your suggestion. The following explanation has been provided: "**Note that 1 nm resolution is equal to 10 $cm^{-1}$ resolution at 1000 nm. The switch from wavelength to wavenumber resolution is done so that the entire spectrum can be covered in a practical number of bands. This wide range allows for complete coverage of stellar and thermal radiative transfer.**"*

- Line 136: Some additional explanation should be provided to explain why species "such as $HNO_4$" are worthy of note.

*Discussion of $HNO_4$ and photolysis initiated by IR is now in Section 4.1.4.*

- Line 141: Sentence beginning " There are hydrocarbon species..." is informal and should be rephrased, e.g. "$H_2O$ and hydrocarbon species (e.g. $CH_4$) also have relevance for early-Earth-like exoplanets"

*This sentence has been changed to **"Species such as $H_2O$ and hydrocarbon species such as $CH_4$ also have relevance for early-Earth-like exoplanets."***

- Line 145: "not suitable" and "can be neglected" seem to be opposites: if Rayleigh scattering by air is a poor model (i.e. not suitable), it should not be used. If the effects of it are so small as to be inconsequential, it can be neglected. The two are not the same. If both somehow apply, this should be further explained.

*Thank for your comment. This has now been reworded-**"It is assumed that most of the absorbed flux from shorter wavelengths is used for dissociation and the atmospheric regions where the absorption occurs will have a significant atomic oxygen and nitrogen content. Rather than formulate a separate scheme that includes O and N scattering we assume absorption will dominate below 175nm and any Rayleigh scattering can be neglected."***

- Line 147: Four sources are listed, not three (Burkholder et al., Atkinson et al., Venot et al., and Hébrard). "Prominent" seems an odd choice of adjective here given one of the sources referred to is private communication – this seems the opposite of prominent.

*The word prominent has been removed.*

- Line 150: "...data was needed and sourced elsewhere." Where the data is from, or reference to where that information is available (Table A1?) should be included in this sentence.

*Thank you for alerting us to this. References have now been provided.*

- Line 153: "...must adjust accordingly via these factors." This sentence does not make sense, should "via" be "due to"?

*This part has been reworded for clarity and now reads as **"The photoelectron factors represent this additional contribution to the effective quantum yield which can then***

*exceed one. This process comes into effect for EUV wavelengths <65 nm and was only required for O₂ of the species considered here."*

- Line 153: "This process was only included for oxygen as the data was available and predominantly comes into effect for EUV wavelengths". Rephrase to clarify the meaning of "only included for O₂ as the data was available" here: the data was only available for O₂ (and not other species), so corrections were only made to O₂, but would ideally be performed for other species; or the correction was applied to O₂ because the data was available, not because it is required as its effect is negligible?

*Thank you for your comment. We have clarified our wording here to:* **"The photoelectron factors represent this additional contribution to the effective quantum yield which can then exceed one. This process comes into effect for EUV wavelengths <65nm and was only required for O₂ of the species considered here."**

- Line 154: EUV should be defined here, if not before

*EUV has now been defined as* **"extreme UV (EUV, <121 nm)"**

- Figure 3: In addition to general points about figures, TOA should be defined, and labels should clarify which model is being used in each case. Does the green line show the Photocomp TOA flux, or is it the Socrates flux at 1 Pa, a value chosen to match the Photocomp TOA?

*The figure captions now clarify which model is being used and provide a clearer definition of TOA.*

- Line 177: Sentence beginning "There they assumed a clear sky..." this sentence is very difficult to read and grammatically incorrect, reword.

*Thank you for this suggestion. This has been reworded for clarity to* **"Part 1a of their experimental set-up was used for the comparisons in this paper. This consists of a clear sky with no aerosols, a Solar Zenith Angle (SZA) of 15˚ over the ocean, an albedo of 0.10 (Lambertian), an incoming Solar irradiance at top-of-atmosphere of 1365Wm⁻² and the inclusion of Rayleigh scattering."**

- Figure 4: In addition to general points about figures, the model being shown in the bottom row should be indicated on the plot.

*Thank you for this recommendation. We have now included details about which models are being used in the caption.*

- Figure 5: In addition to general points about figures, models should be specified on plots and labels should be consistent within a figure. For example, the bottom left panel refers to the green line as "~1 Pa", whereas the bottom right panel refers to the green line as "Photocomp TOA ~1 Pa"

*Thank for your comment. This has now been changed consistently to "~1 Pa".*

- Line 187: For readability, "Sections 4.1.3, 4.1.4, 4.1.5 and 4.1.6 " could be "Sections 4.1.3 – 4.1.6"

*This has been amended to read as* **"Sections 4.1.3 – 4.1.6"**

- Line 207: "Finally, we only present results for the main species in the main part of this paper." This implies the existence of results for minor species elsewhere. I have not been able to find any supplementary information, should this be available somewhere?

*Thank you for pointing this out. This sentence was confusing and misleading and has been omitted. We clarify what species we are specifically looking at and their categories.*

- Line 209: "($^3$P)" should be "O($^3$P)"

*This has now been changed to "O($^3$P)"*

- Page 10 footnote 1 and 2: These links contain spaces and therefore do not work. The data is also not available from the corrected links without permission. If the aim of this paper is to provide an informative comparison of models, it would be useful for this to be generally available if possible, or for the paper to contain instructions on how to request access if not.

*Thank you for alerting us to this. The links no longer have spaces and work. We clarify that the data was provided by Martyn Chipperfield and with his permission.*

- Line 213: "Hartley bands 200-310 nm": Wavelength range should be in brackets

*Brackets have now been included.*

- Line 214: More precise wavelength ranges for the Huggins and Chappuis bands should be provided

*These wavelength ranges have now been defined as **"the Huggins (~300-370 nm) and Chappuis bands (~ 370-790 nm)".***

- Line 220: "...between 220-290 nm, for the lower/mid-atmosphere" there should not be a comma here, and "for" should be "in" instead.

*This sentence has now been amended.*

- Line 226: "...caused by differences in the input data (cross sections, quantum yields and stellar spectrum) between the models." This needs much more detail. Which are different and how do they differ? Are the changes an improvement or not?

*Thank you for your comment. We have now provided more detail about the differences in input data as follows: **"The UCI-ref model is based on JPL recommended cross-sections from 2010 while Socrates is using an updated temperature dependent cross-section from HITRAN 2020 (Gordon et al. 2022). The Socrates model will also have finer sub-band resolution. These differences are likely to be the main cause of the observed discrepancy although without access to the UCI-ref model and data it cannot be reliably determined."***

- Line 228: informal, remove "slightly"

*This word has been removed.*

- Line 234: Sentence beginning "The top left panel of Figure 5...": the greatest disagreement between the rates in Figure 5 occurs at high pressures (> $10^4$ Pa), which is not addressed. What causes this difference?

*Thank you for your comment. This difference has now been addressed and the following explanation has been provided:* **"The divergence in the rates near the surface occurs for values less than 1 x 10$^{-19}$ s$^{-1}$ and is likely due to the use of different O$_2$ cross-sections within the absorption window at wavelengths around 200 nm. There is also a slight departure at very low pressures (~10 Pa, or above ~64 km) where the UCI-ref model is no longer valid as it does not include EUV wavelengths (see discussion in Section 4.1.1)."**

- Line 247: This line refers to small discrepancies due to differences in input data, which are not expanded on in any significant detail either here or in the section referenced.

*Thank you for your comment. We have now provided this explanation:* **"NO$_2$ cross-sections are based on the recommendations of the JPL 19-5 report between 238 nm and 667 nm using data from (Vandaele 1998) at 220 K and 298 K. We use the original high-resolution cross-sections obtained from the MPI-UV/Vis database (Keller 2013) rather than the band means reported in JPL 19-5. We also extend the cross-sections into the FUV and EUV using the data reported in Table A1."**

- Figure 7: In addition to the general points about figures, the line labels in this figure are particularly unclear. The model names are inconsistent across the subplots (SOC vs SOCATES), labels such as "SOC no T QY" and "SOC O$_3$, O$_2$mmr only" are impossible to understand. The pressure levels chosen in the right-hand column change between each subplot and the reason for the choices of levels are not made clear.

*The line labels have all been changed to "Socrates" to be consistent and labels have changed to be clearer. Choices behind the pressure levels presented now have an explanation. For example, for Figure 6, the pressure level at the ozone layer is chosen because of how ozone absorption affects the results*

- Line 250: N($^4$S) is not explained

*An explanation of N($^4$S) "being the ground state of the nitrogen atom" has been added.*

- Line 250: "..as a function of pressure ( Pa, left panels) with UCI-Jxr added (purple line) and as a function of wavelength (µ m) in the right panels." Reorder this sentence to make it clear that UCI-Jxr is added to the left-hand panels to avoid confusion with the magenta line in the right hand panels

*This has been reordered.*

- Line 252: there is clear disagreement in the shape of the profiles below $10^3$ The cause of this should be explained.

*Thank you for your comment. We acknowledge this discrepancy and provide the following explanation:* **"The overall shape of the profile is predominantly governed by the temperature dependent cross-sections which introduce variations that mirror the temperature structure of the atmosphere shown in Figure 1. The Socrates rates are further affected by absorption of the actinic flux at pressures higher than ~ $10^3$ Pa and begin to decrease, whilst the UCI-ref values only show evidence of absorption from ~ 3 x $10^4$ Pa. This difference is likely due to the use of updated ozone absorption cross-sections in Socrates from HITRAN 2020 (see Table A1). We also use high-resolution cross-section data for $NO_2$ which may contribute to the differences as there is fine structure in the near-UV region where photolysis occurs (Akimoto 2016)."**

- Line 255: "The absorption spectrum for $NO_2$ is quite complex". This is vague and imprecise. What is meant by "quite complex"? Presumably that it has fine structure and therefore needs a high resolution spectrum to reproduce the cross section accurately?

*Thank for your comment. This section has been reworded and structured for greater clarity. The reason for the use of high-resolution cross section has been included in the following sentence-* **"We also use high-resolution cross-section data for $NO_2$ which may contribute to the differences as there is fine structure in the near-UV region where photolysis occurs"**

- Line 257: "This is beyond the threshold limit for this photolysis reaction to occur which is 398 nm, but can be accounted for..." This sentence is very difficult to read. Reword the first part of this, e.g. "Beyond the 398 nm threshold for the photolysis reaction to occur..."

*The first part of the sentence has been reworded using your suggestion.*

- Line 258: "This reflects the behaviour of the quantum yield which decreases down to zero between 300-420 nm." Rephrase and/or clarify: is the quantum yield zero from 300 to 420 nm, or does it decrease near 300 to a value of 0 between the two, then increase towards 420 nm? Or does it decrease from a non-zero value at one extreme of the wavelength range to 0 at the other?

*Thank you for your comment. The $NO_3$ section has been reworded for clarity.*

- Line 270: "shows" should be "show".

*This has been fixed.*

- Line 282: "discrepancies lie with different input cross sections and model differences.": This needs to be expanded on. What model differences? Why are different input cross sections used? Are they improvements?

*We were not considering temperature dependence in the cross sections of $N_2O$ before. We have done so now, and our results match the reference results better. This section has been rewritten to reflect this change.*

- Line 283: "The gas NO's photoabsorption cross section" Don't use possessive apostrophes for gases, reword to something like "The photoabsorption cross section of NO gas"

*This has been reworded using your suggestion.*

- Line 288: "Therefore a pressure and temperature dependence, of the absorption coefficients, were included in our calculations." This is not a separate clause and the commas should be removed.

*The commas have been removed.*

- Line 294: "This could indicate a missing component either in the wavelength range included or the quantum yield, or the use of different input data." It seems quite important to know which of these is the case, as one highlights unknowns that need to be addressed in this field of study going forward, whereas the "different input data" could reflect an improvement compared to previous work.

*Thank you for your comment. We clarify in this section exactly what input data we are using as well as highlighting the fact that we use updated ozone cross sections which impact absorption. This section has been reworded to read: **"For HONO we use the recommended cross-section from JPL 19-5 between 184 - 396 nm extended to 400 nm using the 0.5 nm resolution data of (Stutz 2000). The JPL 19-5 cross-section contains a gap between 274 - 296 nm which we fill using the (Stutz 2000) 0.5 nm resolution data between 292 - 296 nm and an interpolation in the logarithm of the cross-sections between 274 - 292 nm. The quantum yield for the reaction HONO -> OH + NO is taken to be unity following the JPL***

*19-5 recommendation. The Socrates photolysis rates shown in Figure 9 are found to be ~ 15% lower than UCI-ref at TOA indicating differences in the HONO cross-section used between the models. Towards higher pressures the Socrates photolysis rates decrease due to absorption of actinic flux in the ozone Huggins bands above 300 nm. We use updated HITRAN 2020 ozone cross-sections in this region compared to UCI-ref which displays less absorption."*

- Line 297: Sentence beginning "For $HNO_3$, we adopt the quantum yield...". This paper would benefit from the quantum yields used being more readily apparent, In cases such as this, where such large changes arise from the choice of quantum yield alone, it would be useful to have the Socrates quantum yields readily available (without having to find the relevant quantum yield in Table A1, find the reference, and find the quantum yield from the reference).

*Thank you for your suggestion. The choice of quantum yields has been clarified in the paper:*
***"Quantum yields for the reaction $HNO_3$ -> $OH$ + $NO_2$ are reported by JPL 19-5 without a specific recommendation so we adopt the recommended values from IUPAC of 0.97 above 248 nm, 0.9 between 200 - 248 nm and 0.33 below 200 nm."***

- Line 299: "...significant discrepancy between the Socrates and reference rates, unless we adopt a quantum yield of one and the near–infrared cross sections and quantum yields are omitted." This should be clarified: In all of these cases, are the reference models wrong and they were missing factors (variable quantum yields, IR cross sections etc.) and this work is a clear improvement due to the inclusion of these things, or are the expected quantum yields adopted in this work in some way uncertain or untrustworthy?

*We have expanded upon this section and provided further detail. We failed to include temperature dependence cross-sections for 200K and 220K for $HNO_3$. This has been considered in the results now. With regards to the quantum yield we now provide the following explanation behind our choice and why it may differ from what the reference model used:*

***"Quantum yields for the reaction $HNO_3$ -> $OH$ + $NO_2$ are reported by JPL 19-5 without a specific recommendation so we adopt the recommended values from IUPAC of 0.97 above 248 nm, 0.9 between 200 - 248\,nm and 0.33 below 200 nm. The resultant photolysis rates from Socrates in Figure 10 are significantly lower than those of UCI-ref. We note that a quantum yield of 1 for wavelengths >200 nm is consistent with the range of reported values in JPL 19-5. If we use a quantum yield of 1 for all wavelengths (blue dashed line) the photolysis rates match the UCI-ref values very well."***

*For $HO_2NO_2$ we now provide a clearer analysis:* ***"For this comparison we ignore photodissociation in the overtone and combination bands in the infra-red. For the reaction $HO_2NO_2$ -> $OH$ + $NO_3$ we use the JPL 19-5 recommended quantum yields of 0.3 at***

*wavelengths $<200\,nm$ and 0.2 >200 nm. The photolysis rates shown in Figure 10are significantly lower than UCI-ref. However, if we again use a quantum yield of 1 at all wavelengths (blue dashed line) the rates match the UCI-ref values reasonably well, indicating the reference results may actually be for the total $HO_2NO_2$ photolysis rate rather than the particular channel specified for PhotoComp."*

- Line 302: $S(^3P)$ is not explained (ground state vs excited)

*An explanation of the ground state has been provided.*

- Line 307: "mid atmosphere" should be "mid-atmosphere"

*This has been changed so that mid-atmosphere is consistent throughout the paper.*

- Line 309: Sentence beginning "The main difference in our rates and those of...": Is this difference in choice of input data an improvement (for example, using higher resolution data, capturing the fine structure of the cross-section) or a limitation (for example, lack of pressure and temperature dependence)?

*Thank you for your suggestion. Our previous explanation was not accurate and the differences are a result of the pressure dependence not being considered. We provide the following explanation: "…the quantum yield has a strong dependence on both pressure and temperature which we have not taken into account as the functionality to incorporate a pressure dependence has not yet been included in Socrates."*

- Line 313: carbonyl sulphide should not be capitalised

*This has been amended.*

- Line 314: "... which implies that the reference also included this." This sentence suggests that part of the aim of this paper is to critique the reference models, back-out their assumptions and choice of values. If this is an aim of the paper, it should be mentioned earlier and receive more consideration throughout the paper. If not, what is the cause of the agreement worthy of note? Presumably the most important this is that the correct values are used going forward?

*This is not one of the aims of the paper. This sentence has been omitted and we simply state exactly what input data we use.*

- Line 326; "Glyoxal" should not be capitalised

*This has been amended.*

- Line 328: "The rates calculated by Socrates $CH_3ONO_2$ are also significantly..." Something is missing to make this sentence make sense, possibly the word "for"?

*The missing 'for' has been included.*

- Line 342: "This is an appropriate total incoming flux for a planet in the habitable zone around Prox Cen." Consider adding further information here about corrections to the flux for planets at different distances from their stars. Are the values used here still valid if the total incoming flux is rescaled? What corrections would be necessary in that case?

*Thank you for your suggestion. We have provided extra detail:* **"However, we maintain the same total TOA incoming flux at 1365Wm⁻² as used for the Solar calculations in Section 4.1 to make comparison between the resulting rates easier. This is an appropriate total incoming flux for a planet in the habitable zone around Proxima Centauri. Note that for planets at different orbital distances the TOA flux will change according to the inverse square law, while the photolysis rates will scale linearly with the TOA flux."**

- Line 344: "FUV" must be defined here, if not before

*This has been defined earlier in the paper.*

- Line 345: "This has implications for the photolysis rates of certain species..." List the species or refer to the sections where this will be discussed further

*This sentence has been omitted and where relevant, this is discussed further along in the paper.*

- Line 349: "Figure 13 shows the actinic flux at three different atmospheric pressure levels, namely the TOA (blue)..." Presumably this section uses an identical atmosphere to the

Earth's, but placed around Proxima Centauri? This should be explicitly stated somewhere in the introduction to Section 4.2 to make this clear.

*Thank you for your comment. In the introduction to Section 4.2 we have included the sentence:* **"In this work, we perform calculations using Socrates with the same Earth-like atmosphere described in Section 4.1.1 but with the Solar irradiation replaced with the irradiation of an M dwarf. This provides a set of initial benchmark rates for the major species and photolysis reactions."** *, to make this clearer.*

- Line 356: "...on a log scale as a function of pressure (Pa, top panels) and wavelength (μm, bottom panels) for just the Prox Cen spectrum." The figure contains rates as a function of pressure for both Proxima Centauri and the Sun in the top panels. The wording of this sentence currently implies the figure shows only data for Proxima Centauri, so should be corrected to make it clear when both stars are plotted.

*Thank you for alerting us to this. This has been corrected to make it clear that both stars are plotted.*

- Line 358: "...are much lower than Solar because the visible part contributes a larger proportion of the total rate and there is much less NUV when compared with the Solar spectrum results..." Some additional explanation should be included here to explain this point better, especially as the photolysis cross sections of species at all wavelengths (i.e., without the effect of the stellar spectra included) are not available in this paper. Are the photolysis cross sections lower in the visible range compared to the near-UV, and Proxima Centauri has more power in the visible range and less in the near-UV compared to the Sun, meaning photolysis is less efficient for Proxima Centauri?

*Thank you for this suggestion. We have expanded upon this point, and the explanation now reads as follows:*

**"For ozone, $O_3$, the Proxima Centauri photolysis rates are significantly lower than the Solar case due to the lower stellar irradiance in the region 200 - 300 nm coinciding with the strong Hartley absorption bands of ozone (compare Figures 2 and 13). Proxima Centauri is a much cooler star than the Sun with a spectrum that peaks further towards the red, with significantly more power in the visible than the near-UV. For the $O_3 \rightarrow O_2 + O(^3P)$ reaction this means there is a larger relative contribution from the weak Chappuis absorption bands beyond 400 nm than from the Hartley bands. These weak bands do not have a significant effect on the actinic flux in the visible region and as a result the photolysis rates do not experience the sharp decline across the ozone layer that is seen with the Solar case."**

- Line 359: "figure 4" should be capitalised

*This has been amended.*

- Line 368: "the major contribution is from wavelengths > 175nm". Some explanation here of why the Lyman-α line is not the dominant contribution despite having a photolysis rate three orders of magnitude higher in Figure 15 (bottom right) than values above 175 nm.

*Thank you for pointing this out. We have now provided an explanation***: "For the Solar case, the major contribution is from the Schumann Runge absorption bands at wavelengths > 175 nm while for Proxima Centauri there is a much larger contribution from Lyman- α wavelengths. This leads to approximately equal total photolysis rates for $O_2$ -> $O(^3P)$ + $O(^3P)$ at TOA. However the rates for Proxima Centauri decrease much more rapidly towards higher pressures due to stronger attenuation of Lyman-α wavelengths."**

- Line 382: This should presumable by "~80-130 nm", not "~0.08-130 nm"

*This has been amended.*

- Line 394: "For the Prox Cen case oxygen absorption..." requires a comma after "case"

*A comma has been added.*

- Line 400: Comparisons between the Solar and Proxima Centauri spectra would be easier if they were included within the same figure, as is the case for many other reactions.

*Figure 17 has been amended to include both the Solar and Proxima Centauri spectra.*

- Line 402: Sentence beginning "As there is very little absorption..." "different pressures is almost entirely" should be "different pressures are almost entirely"

*Thank you. This has now been changed.*

- Line 407: Sentence beginning 2"As the Prox Cen spectrum..." The large number of clauses in this sentence makes it difficult to read. Consider re-ordering to something like "As it is a lower temperature star, Proxima Centauri has a higher fraction of its flux at longer wavelengths, and rates are therefore sensitive to the temperature of the atmosphere."

*Thank you for your suggestion. This has been reworded to read **"As it is a lower temperature star, Proxima Centauri has a higher fraction of its flux at longer wavelengths and rates are therefore increased as the temperature increases."***

- Line 414: Sentence beginning "Figure 20 shows rates..." Some reactions are written out in words ($H_2CO$ into H and HCO), others are written as reactions ($H_2CO \rightarrow H_2 + CO$). This should be broadly consistent throughout the paper, but must be consistent within a sentence.

*Throughout the paper, all reactions are now consistently written as reactions.*

- Line 414: Sentence beginning "Figure 20 shows rates...": "top row", "middle row", "bottom row", "left column", "middle column", and "right column" must be in brackets

*Brackets have now been included.*

- Line 425: "top row" should be in brackets

*Brackets have now been included.*

- Line 425: "4" would be better written as "four" here

*This has been changed.*

- Line 426: "the methyl radical-$CH_3$" does not make sense and should be either "$CH_3$", "the methyl radical", or ideally "the methyl radical, $CH_3$,", or "the methyl radical ($CH_3$)"

*This has been changed to **"the methyl radical, $CH_3$,".***

- Line 427: "0.2155 nm" should be either 0.2155 µm or 215.5 nm.

*This was an error. It has been changed to "215.5 nm".*

- Line 456: Sentence beginning "Whereas the photolysis rates..." This is not a full sentence and should be expanded upon, reworded, or combined with the previous sentence.

*This sentence has been reworded.*

- Figure 12: Including the name of the star in the legend would make this plot much easier to read (e.g. "Proxima Centauri (MUSCLES-Ribas)" and "Solar (CMIP6)")

*Thank you for your suggestion. All legends have been changed to include the name of the star.*

- Figure 12: CMIP6/cmip6 should be consistent across the figure and description

*All instances have been changed to CMIP6 for consistency.*

- Figure 12: provide a value for the total incoming flux, e.g. "The value of 0.02 AU was selected to provide a total incoming flux of [value], consistent with the Solar spectrum."

*The total incoming flux of 1365 Wm$^{-2}$ has been included in the caption.*

- Figure 13: Some sort of labelling of which is the solar spectrum, and which is Proxima Centauri on the plot itself would improve readability immensely.

*Thank you for your suggestion. The legend labels now provide information which is the solar spectrum, and which is Proxima Centauri*

- Figure 14 (and onwards): Presumably the solar case results here are for Socrates? This should be explicitly stated as numerous models for the solar case have been used in this paper

*Thank you. We now make it explicitly clear that these results are yielded by Socrates.*

- Figure 16 (and onwards): subplot titles or captions on the figures must be included here to identify what reactions, stars, and models each subplot is showing.

*All captions for Figure 16 and onwards have been adjusted to provide more useful information to describe the subplots.*

- Figure 18: For clarity, could these be shown as insets in the respective subplots of Figure 17? Could similar zoomed-in sections be shown for the solar case (on a secondary axis/scaled to comparable values)?

*Thank you for your suggestion. The results for the Solar case are displayed on a linear scale in the previous Section (Section 4.1.5). We found that displaying them separately allowed for the graphs to be read better.*

- Table A1: Caption should be before the table

*The caption is now before the table.*

- Table A1: Second words of two-word species should not be capitalised (Hydrogen Peroxide, Nitrogen Dioxide, Nitrous Oxide, Nitric Oxide, Nitrous Acid, Nitric Acid, Carbon Dioxide)

*Two word species are no longer capitalized.*

- Table A1: $OH(X_2\pi)$ is written as $OH(X^2\Pi)$ in the paper. This should be consistent.

*This has been changed to $OH(X^2\Pi)$ consistently throughout the paper.*